# Fast and Accurate Blind Flexible Docking

**Zizhuo Zhang**[1]  **Lijun Wu**[2†]  **Kaiyuan Gao**[3]  **Jiangchao Yao**[4]  **Tao Qin**[5]  **Bo Han**[1†]

[1]TMLR Group, Department of Computer Science, Hong Kong Baptist University
[2]Shanghai AI Laboratory    [3]Huazhong University of Science and Technology
[4]CMIC, Shanghai Jiao Tong University    [5]Microsoft Research AI4Science
{cszzzhang, bhanml}@comp.hkbu.edu.hk, apeterswu@gmail.com
im_kai@hust.edu.cn, Sunarker@sjtu.edu.cn, taoqin@microsoft.com

## Abstract

Molecular docking that predicts the bound structures of small molecules (ligands) to their protein targets, plays a vital role in drug discovery. However, existing docking methods often face limitations: they either overlook crucial structural changes by assuming protein rigidity or suffer from low computational efficiency due to their reliance on generative models for structure sampling. To address these challenges, we propose FABFlex, a fast and accurate regression-based multi-task learning model designed for realistic blind flexible docking scenarios, where proteins exhibit flexibility and binding pocket sites are unknown (blind). Specifically, FABFlex's architecture comprises three specialized modules working in concert: (1) A pocket prediction module that identifies potential binding sites, addressing the challenges inherent in blind docking scenarios. (2) A ligand docking module that predicts the bound (holo) structures of ligands from their unbound (apo) states. (3) A pocket docking module that forecasts the holo structures of protein pockets from their apo conformations. Notably, FABFlex incorporates an iterative update mechanism that serves as a conduit between the ligand and pocket docking modules, enabling continuous structural refinements. This approach effectively integrates the three subtasks of blind flexible docking—pocket identification, ligand conformation prediction, and protein flexibility modeling—into a unified, coherent framework. Extensive experiments on public benchmark datasets demonstrate that FABFlex not only achieves superior effectiveness in predicting accurate binding modes but also exhibits a significant speed advantage (208×) compared to existing state-of-the-art methods. Our code is released at https://github.com/tmlr-group/FABFlex.

## 1 Introduction

Molecular docking is a pivotal technology in drug discovery, aiming at predicting the binding structures of ligand-protein complexes to elucidate how drug-like small molecules (ligands) interact with target proteins (Morris et al., 1996; Morris & Lim-Wilby, 2008; Agarwal & Mehrotra, 2016). Over the past decades, the development of molecular docking has seen continuous breakthroughs, evolving from traditional simulation software grounded in the principles of physics and chemistry to recent deep learning-based methods (Crampon et al., 2022). Traditional methods typically utilize empirical energy scoring functions to rank numerous searched conformations (Trott & Olson, 2010; Friesner et al., 2004; McNutt et al., 2021; Koes et al., 2013), often resulting in excessive processing times and heavy computational burdens.

The shift towards deep learning-based docking approaches represents a significant transformation in the field, offering an alternative pathway for exploring protein-molecule interactions. Despite technological advances, a substantial number of these methods (Lu et al., 2022; Zhang et al., 2023; Stärk et al., 2022; Zhou et al.; Pei et al., 2024; Gao et al., 2024; Corso et al., 2023) rely on the *rigid docking* paradigm, which assumes that proteins are rigid and remain static during the docking process. This simplification contradicts the physiological reality of molecular docking, where proteins

---

†Correspondence to Bo Han (bhanml@comp.hkbu.edu.hk) and Lijun Wu (apeterswu@gmail.com).

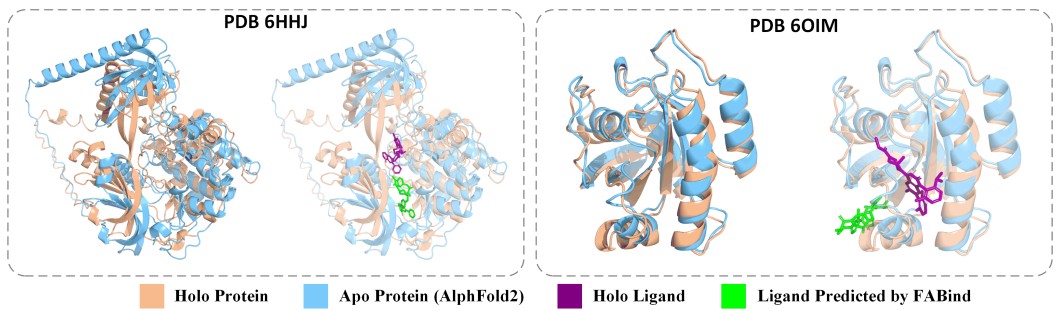

Figure 1: The two cases illustrate our motivation. These two cases, involving PDB 6HHJ and PDB 6OIM, highlight the structural discrepancy between apo proteins (AlphaFold2) and holo proteins. In the two cases, existing rigid docking method FABind (Pei et al., 2024) yields incorrect molecular docking results when apo proteins are inputted as direct substitutes for the original holo proteins.

exhibit significant flexibility and dynamic behavior (Henzler-Wildman & Kern, 2007; Lane, 2023; Popovych et al., 2006), thereby restricting the practical applicability of these methods in real-world scenarios. Consequently, it is imperative to explore and develop *flexible docking* methods that more faithfully reflect the realistic docking process (Sahu et al., 2024).

The advent of AlphaFold series (Senior et al., 2020; Jumper et al., 2021; Abramson et al., 2024) has revolutionized the technology of protein structure prediction, enabling precise 3D protein structure predictions without relying on laborious methods such as X-ray crystallography (Ilari & Savino, 2008), or cryo-electron microscopy (Bonomi & Vendruscolo, 2019). Yet, integrating AlphaFold-predicted apo structures into existing docking workflows remains challenging. For instance, as two cases shown in Fig. 1, a notable discrepancy often exists between the AlphaFold2-predicted apo structures (Jumper et al., 2021) and the actual docked holo structure. Moreover, existing rigid docking methods (Lu et al., 2022; Pei et al., 2024) often yield wrong docking results when the AlphaFold2-predicted apo proteins are used as direct substitutes for previously used holo protein.

Recent flexible docking methods (Lu et al., 2024; Huang et al.; Zhang et al., 2024; Qiao et al., 2024) predominantly rely on diffusion models and sampling strategy (Yang et al., 2023) due to the strong distribution modeling capability of generative models. For example, DynamicBind (Lu et al., 2024) utilizes equivariant geometric diffusion networks with multiple sampling iterations to reconstruct holo protein structures from AlphaFold2-based conformations. Though effective, the step-by-step diffusion process and extensive sampling requirements inherently reduce computational efficiency. In contrast, regression-based methods (Lu et al., 2022; Zhang et al., 2023; Stärk et al., 2022), offer a faster alternative by directly predicting the bound structures of protein-ligand complexes using well-designed neural networks. Notably, regression-based approaches have not yet been explored for flexible docking scenarios. Within this category, the FABind series (Pei et al., 2024; Gao et al., 2024) is committed to balance docking accuracy with computational efficiency through a multi-task model. However, the FABind series still adheres to the rigid protein assumption, limiting its efficacy in realistic flexible docking scenario. Consequently, the challenge of developing a method that combines both accuracy and speed for blind flexible docking remains largely unexplored.

In this study, we propose FABFlex, a regression-based multi-task learning model crafted to achieve both Fast and Accurate Blind Flexible docking. Specifically, FABFlex consists of three designed modules, each targeting a subtask decomposed from the process of blind flexible docking: (1) a *pocket prediction module* that identifies binding pocket sites; (2) a *ligand docking module* that predicts the structure of holo ligand; (3) a *pocket docking module* that predicts the structure of holo pocket. Each module is built using an E(3)-equivariant graph neural network layer, called "FABind layer" (Pei et al., 2024), which is tailored to handle the ligand-protein heterogeneous graph. The operational pipeline of FABFlex begins with the pocket prediction module, which determines the protein residues constituting the binding pocket, effectively navigating the challenge of the blind setting. Subsequently, the ligand and pocket docking modules leverage the predicted pocket information to predict the holo structures of the ligand and pocket, respectively. An iterative update mechanism facilitates the exchange of structural predictions between the ligand and pocket docking modules, allowing for further coordinate refinements. Collectively, these three modules function in

a seamless end-to-end process, allowing FABFlex to simultaneously predict the holo structures of both the ligand and pocket in a single operation, thus ensuring faster computation.

We conduct experiments on public docking benchmark PDBBind to evaluate our FABFlex against a variety of docking methods. Compared to the baselines, FABFlex effectively increases the percentage of ligand RMSD below 2Å to 40.59% and reduces the pocket RMSD to 1.10Å. Notably, FABFlex significantly accelerates the computational speed, approximately 208 times faster than the recent state-of-the-art flexible docking method, DynamicBind (Lu et al., 2024).

## 2 RELATED WORK

**Molecular docking.** As a cornerstone of drug discovery, molecular docking, often synonoymous with ligand-protein docking, focuses on the interactions between ligands and proteins. Traditional methods like Vina (Trott & Olson, 2010), Smina (Koes et al., 2013), Glide (Friesner et al., 2004), Gnina (McNutt et al., 2021) and Gold (Jones et al., 1997), use physics-based scores to analyze these interaction, which, though effective, tend to be computationally intensive. Recent progress in geometric deep learning has sparked the development of deep learning-based docking strategies (Crampon et al., 2022), which can be broadly categorized into regression-based and sampling-based methods. Regression-based methods like EquiBind (Stärk et al., 2022), TankBind (Lu et al., 2022), E3Bind (Zhang et al., 2023) and FABind (Pei et al., 2024) leverage various geometric neural networks to directly predict binding structures. Conversely, sampling-based methods like Diff-Dock (Corso et al., 2023), manipulate the rotation, translation and torsion of ligands using diffusion models. These methods generally simplify the problem by assuming protein rigidity, neglecting the dynamic nature of protein in realistic docking scenarios.

**Flexible molecular docking.** Recent methods in flexible docking, such as DynamicBind (Lu et al., 2024), ReDock (Huang et al.), PackDock (Zhang et al., 2024) and NeuralPLexer (Qiao et al., 2024), are primarily based on diffusion models with sampling strategy. For example, DynamicBind (Lu et al., 2024) uses equivariant geometric diffusion networks to reconstruct the holo structures of both ligand and protein from their apo states. ReDock (Huang et al.) adopts the neural diffusion bridge model that employs energy-to-geometry mapping on geometric manifolds to predict protein-ligand binding structures. While these methods effectively enhance the docking performance, they suffer from the typical flaws associated with diffusion models and multi-round sampling strategy, i.e., low computational efficiency. This limitation impedes the scalability of these methods in assessing extensive volumes of potential, unknown molecule-protein interactions, which are crucial for advancing drug discovery. In this paper, we attempt to provide a regression-based solution aimed at achieving both fast inference and high docking accuracy.

## 3 METHODOLOGY

### 3.1 PROBLEM STATEMENT AND PRELIMINARY

**Notations.** Each ligand-protein complex is denoted as a heterogeneous graph $\mathcal{G} = \{\mathcal{V} := (\mathcal{V}^l, \mathcal{V}^p), \mathcal{E} := (\mathcal{E}^l, \mathcal{E}^p, \mathcal{E}^{lp})\}$, where $\mathcal{V}$ and $\mathcal{E}$ denote the sets of nodes and edges, respectively. Specifically, in the ligand subgraph $\mathcal{G}^l = \{\mathcal{V}^l, \mathcal{E}^l\}$, each node $v_i = (\mathbf{h}_i, \mathbf{x}_i) \in \mathcal{V}^l$ corresponds an atom of ligand, with $\mathbf{h}_i \in \mathbb{R}^{d^l}$ representing the pre-extrated feature by TorchDrug (Zhu et al., 2022) and $\mathbf{x}_i \in \mathbb{R}^3$ specifying its spatial coordinate. The edge set $\mathcal{E}^l$ encompasses the chemical bonds within the ligand. In the protein subgraph $\mathcal{G}^p = \{\mathcal{V}^p, \mathcal{E}^p\}$, each node $v_j = (\mathbf{h}_j, \mathbf{x}_j) \in \mathcal{V}^p$ represents a residues, with $\mathbf{h}_j \in \mathbb{R}^{d^p}$ derived from ESM-2 (Lin et al., 2022) as the pretrained feature, and $\mathbf{x}_j \in \mathbb{R}^3$ indicating the coordinate of the $C_\alpha$ atom in the residue. The edge set $\mathcal{E}^p$ connects residues that are within an 8Å distance. Additionally, the set of external interface edges, denoted as $\mathcal{E}^{lp}$, comprises edges that connect nodes from the ligand set $\mathcal{V}^l$ to the protein set $\mathcal{V}^p$ when they are spatially within 10Å of each other. When specifically focusing on the true pocket region, the ligand-pocket complex is depicted as a reduced heterogeneous graph $\mathcal{G}^* = \{\mathcal{V}^* := (\mathcal{V}^l, \mathcal{V}^{p*}), \mathcal{E}^* := (\mathcal{E}^l, \mathcal{E}^{p*}, \mathcal{E}^{lp*})\}$, where $\mathcal{V}^{p*}, \mathcal{E}^{p*}$ and $\mathcal{E}^{lp*}$ are derived from the subsets of the protein that constitutes the pocket. Similarly, a hat is used in the notation to represent the predicted pocket, $\hat{\mathcal{G}}^* = \{\hat{\mathcal{V}}^* := (\mathcal{V}^l, \hat{\mathcal{V}}^{p*}), \hat{\mathcal{E}}^* := (\mathcal{E}^l, \hat{\mathcal{E}}^{p*}, \hat{\mathcal{E}}^{lp*})\}$.

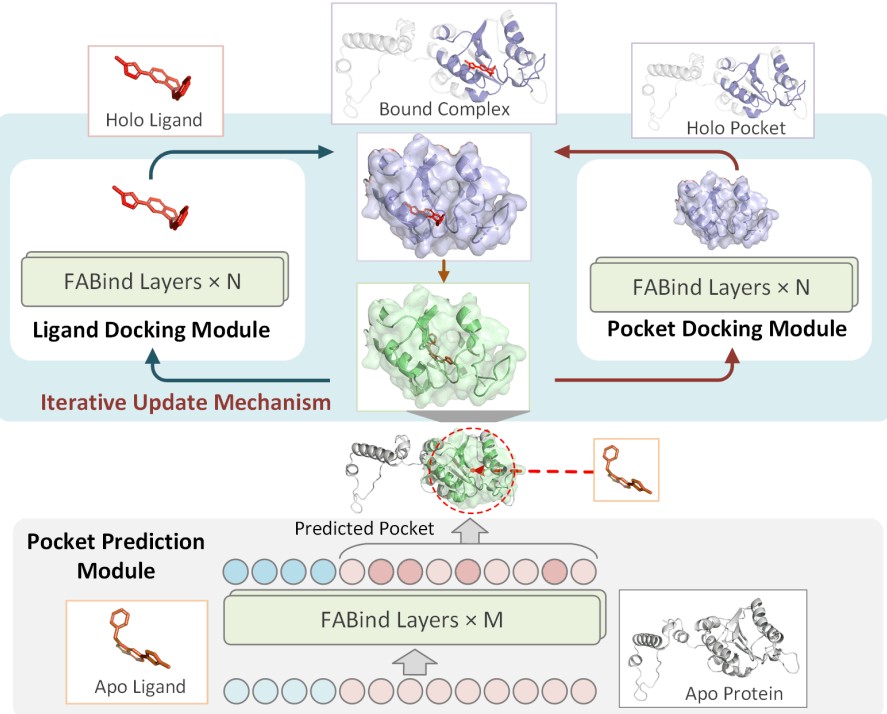

Figure 2: The overview of proposed FABFlex model, which consists of a *pocket prediction module*, a *ligand docking module*, and a *pocket docking module*. The pocket prediction module identifies pocket residues within the protein. Based on the predicted binding pocket sites, the ligand docking module and pocket docking module predict the holo structures of the ligand and pocket, respectively. An *iterative update mechanism* facilitates the exchange of predictions between the ligand and pocket docking modules, enabling further coordinate refinements. These modules work together within a unified end-to-end model for the blind flexible docking scenario, i.e., "(apo protein, apo ligand) → (holo pocket, holo ligand)".

**Blind Flexible Docking.** Given an unbounded pair of conformations: an *apo ligand* randomly initialized by RDKit (Landrum et al., 2013) and an *apo protein* predicted by AlphaFold2 (Jumper et al., 2021), the aim of blind flexible molecular docking is to predict the bound structure of ligand-pocket complex, i.e., *holo ligand* and *holo pocket*, denoted as $\hat{\mathbf{x}} = \{\{\hat{\mathbf{x}}_i\}_{1 \leq i \leq n^l}, \{\hat{\mathbf{x}}_j\}_{1 \leq j \leq n^{p*}}\}$, where $n^l = |\mathcal{V}^l|$ and $n^{p*} = |\mathcal{V}^{p*}|$ indicate the number of ligand atoms and pocket residues respectively. Unlike most existing studies that assume protein rigidity (Pei et al., 2024) or rely on pre-known pocket sites (Zhang et al., 2024), the blind flexible setting reflects the challenges encountered in real-world molecular docking scenarios.

**Fundamental Component: FABind Layer.** FABind (Pei et al., 2024) and its enhanced version, FABind+ (Gao et al., 2024), establish an end-to-end deep learning framework that simultaneously predicts binding pocket sites and holo ligand structure under rigid setting. The fundamental geometric graph neural network used in both two approaches is called "FABind layer", denoted as $\mathfrak{F}(\cdot)$, which is an improved E(3)-equivariant graph neural networks (EGNN) (Satorras et al., 2021) tailored for ligand-protein complex graph. The $l$-th FABind layer is denoted as follows:

$$\mathbf{h}_i^{(l+1)}, \mathbf{h}_j^{(l+1)}, \mathbf{x}_i^{(l+1)}, \mathbf{x}_j^{(l+1)}, \mathbf{p}_{ij}^{(l+1)} = \mathfrak{F}(\mathbf{h}_i^{(l)}, \mathbf{h}_j^{(l)}, \mathbf{x}_i^{(l)}, \mathbf{x}_j^{(l)}, \mathbf{p}_{ij}^{(l)}), \quad (1)$$

where $p_{ij} \in \mathbb{R}^d$ is the pair embedding of the ligand-protein node pair $(v_i, v_j) \in \mathcal{V}^l \times \mathcal{V}^p$, and $d$ is the hidden size. Similar to general graph neural networks (Wu et al., 2020), we can stack multiple FABind layers to extract deeper features and capture high-order information within the graph. Inspired by the fast inference of the FABind series methods, we employ the FABind layer as our fundamental component in constructing each module of our FABFlex.

## 3.2 FABFlex

### 3.2.1 Design Philosophy

The design philosophy of FABFlex centers on achieving both high efficiency and accuracy in blind flexible docking task, which can be decomposed into three key subtasks: identifying pocket sites, predicting the bound holo structure of ligand, and predicting the bound holo structure of pocket. Specifically, the first subtask can be modelled as a binary classification problem to determine which protein residues form the docking pocket, while the other two subtasks are 3D coordinate regression problems. To pursue faster docking computation, the FABFlex model is designed to predict the docking results in a single-pass operation, without requirements of extensive sampling and repetitive computations. Additionally, FABFlex aims to operate without relying on external tools that could add extra computational overhead, such as the pocket detection tool P2Rank (Krivák & Hoksza, 2018), which is used in existing studies (Lu et al., 2022; Zhang et al., 2023) to detect candidate pocket sites. According to above philosophy, the FABFlex model builds on an end-to-end multi-task learning framework tailored for blind flexible docking.

### 3.2.2 Architecture of FABFlex

An overview of the proposed FABFlex model is illustrated in Fig. 2. Designed to tackle the intricacy of blind flexible docking, FABFlex employs a collaborative architecture consisting of three specialized modules: (1) a *pocket prediction module*, denoted as $\mathcal{M}_S(\cdot)$; (2) a *ligand docking module*, denoted as $\mathcal{M}_L(\cdot)$; (3) a *pocket docking module*, denoted as $\mathcal{M}_P(\cdot)$. All modules are implemented using stacked FABind layers, while each tailored to a specific subtask decomposed from blind flexible docking. Specifically, the pocket prediction module is responsible for identifying the residues that form the pocket, addressing the "blind" issue, as formulated below:

$$
\begin{aligned}
\{\hat{y}_j\}_{1\leq j\leq n^p} &= \mathcal{M}_S(\mathcal{G}, \{\mathbf{h}_i, \mathbf{x}_i\}_{1\leq i\leq n^l}, \{\mathbf{h}_j, \mathbf{x}_j\}_{1\leq j\leq n^p}), \\
\{v_j\}_{1\leq j\leq \hat{n}^{p*}} &= \{\hat{y}_j \odot v_j\}_{1\leq j\leq n^p} \in \hat{\mathcal{V}}^{p*},
\end{aligned}
\tag{2}
$$

where $\mathcal{G}$ is the ligand-protein graph, $\{\hat{y}_j\}_{1\leq j\leq n^p}(\hat{y}_j \in \{0,1\})$ is an indicator vector predicted by the pocket prediction module to locate the pocket sites, $\hat{\mathcal{V}}^{p*}$ denotes the predicted set of pocket residues and $\hat{n}^{p*} = |\hat{\mathcal{V}}^{p*}|$, $\odot$ symbolizes the selection operation based on the indicator vector. The predicted pocket sites by pocket prediction module enable the subsequent ligand and pocket docking modules to concentrate effectively on the crucial pocket area, narrowing the large ligand-protein graph to a more targeted ligand-pocket graph.

The ligand docking module and the pocket docking module cater to the "flexible docking" by predicting the bound structures of holo ligand and holo pocket respectively, formulated as follows:

$$
\begin{aligned}
\{\hat{\mathbf{x}}_i\}_{1\leq i\leq n^l} &= \mathcal{M}_L(\hat{\mathcal{G}}^*, \{\mathbf{h}_i, \mathbf{x}_i\}_{1\leq i\leq n^l}, \{\mathbf{h}_j, \mathbf{x}_j\}_{1\leq j\leq \hat{n}^{p*}}), \\
\{\hat{\mathbf{x}}_j\}_{1\leq j\leq \hat{n}^{p*}} &= \mathcal{M}_P(\hat{\mathcal{G}}^*, \{\mathbf{h}_i, \mathbf{x}_i\}_{1\leq i\leq n^l}, \{\mathbf{h}_j, \mathbf{x}_j\}_{1\leq j\leq \hat{n}^{p*}}),
\end{aligned}
\tag{3}
$$

where $\hat{\mathcal{G}}^*$ is the ligand-pocket graph, $\{\hat{\mathbf{x}}_i\}_{1\leq i\leq n^l}$ and $\{\hat{\mathbf{x}}_j\}_{1\leq j\leq \hat{n}^{p*}}$ are predicted structures of ligand and pocket respectively. In current model, the ligand docking module and the pocket docking module predict the structures in isolation, which has a gap to reflect the interactive influence between ligand atoms and pocket residues in docking process. To rectify this, there is a need for a bridge that connects the ligand docking module and pocket docking module, enhancing the model to capture the nature of interactive dynamics of docking.

### 3.2.3 Iterative Update Mechanism

To facilitate the exchange of predicted structures between the ligand docking module and the pocket docking module, we introduce an iterative update mechanism to further promote the coordinate refinement. Specifically, the predicted ligand and pocket, which together form the updated ligand-pocket graph, are fed back into their respective modules to produce new predictions. This $k$-th iterative process can be formulated as follows:

$$
\begin{aligned}
\{\hat{\mathbf{x}}_i^{(k+1)}\}_{1\leq i\leq n^l} &= \mathcal{M}_L(\hat{\mathcal{G}}^{(k)*}, \{\mathbf{h}_i, \hat{\mathbf{x}}_i^{(k)}\}_{1\leq i\leq n^l}, \{\mathbf{h}_j, \hat{\mathbf{x}}_j^{(k)}\}_{1\leq j\leq \hat{n}^{p*}}), \\
\{\hat{\mathbf{x}}_j^{(k+1)}\}_{1\leq j\leq \hat{n}^{p*}} &= \mathcal{M}_P(\hat{\mathcal{G}}^{(k)*}, \{\mathbf{h}_i, \hat{\mathbf{x}}_i^{(k)}\}_{1\leq i\leq n^l}, \{\mathbf{h}_j, \hat{\mathbf{x}}_j^{(k)}\}_{1\leq j\leq \hat{n}^{p*}}),
\end{aligned}
\tag{4}
$$

where $\mathcal{G}^{(k)*}$ denotes the ligand-pocket graph updated by the $k$-th iteration's predicted coordinates of ligand atoms $\{\hat{\mathbf{x}}_i^{(k)}\}_{1 \le i \le n^l}$ and pocket residues $\{\hat{\mathbf{x}}_j^{(k)}\}_{1 \le j \le \hat{n}^{p*}}$, and the number of iterations $K$ is a hyperparameter discussed in Appendix C.5. Notably, the update mechanism iterates solely on the coordinates, excluding the features. Additionally, inspired by AlpahFold2 (Jumper et al., 2021), we only record the gradient at the final iteration during training process. These strategies are beneficial in reducing the memory demands and easing computational burden.

### 3.2.4 TRAINING LOSS AND PIPELINE

**Training Loss.** FABFlex is built on a multi-task learning framework, utilizing multiple losses to supervise different modules from various aspects. Referring to FABind series (Pei et al., 2024; Gao et al., 2024), the training loss comprises pocket prediction loss, ligand coordinate loss, pocket coordinate loss, and distance map constraint loss, formulated as follows:

$$\mathcal{L} = \alpha_1 \mathcal{L}_{\text{pocket\_pred}} + \alpha_2 \mathcal{L}_{\text{ligand\_coord}} + \alpha_3 \mathcal{L}_{\text{pocket\_coord}} + \alpha_4 \mathcal{L}_{\text{dis\_map}}, \tag{5}$$

where the pocket prediction loss $\mathcal{L}_{\text{pocket\_pred}}$ encompasses a residue classification loss and a pocket center loss. The ligand coordinate loss $\mathcal{L}_{\text{ligand\_coord}}$ is a Huber regression loss to measure the distance between predicted and ground-truth coordinates of ligand atoms. Similarly, the pocket coordinate loss $\mathcal{L}_{\text{pocket\_coord}}$ is a Huber loss to compute the coordinate distance between predicted and ground-truth pocket residues. The distance map loss $\mathcal{L}_{\text{dis\_map}}$ supervises the predicted relative distance of atom-residue pairs. The details of implement of training loss are provided in Appendix A.1.

**Pipeline.** Given an apo ligand initilized by RDKit and an apo protein predicted by AlphaFold2, the ligand is initially positioned at the center of the protein to construct the ligand-protein graph $\mathcal{G}$. This graph is passed through the pocket prediction module to identify the binding pocket residues $\{v_j\}_{1 \le j \le \hat{n}^{p*}}$. Navigated by the predicted pocket, the ligand is translated from the protein center to the pocket center, creating the ligand-pocket graph $\hat{\mathcal{G}}^*$. This graph is then fed into the ligand and pocket docking modules, where it undergoes refinement through an iterative update mechanism. At the final iteration, the predicted holo structures of ligand $\{\hat{\mathbf{x}}_i\}_{1 \le i \le n^l}$ and pocket $\{\hat{\mathbf{x}}_j\}_{1 \le j \le \hat{n}^{p*}}$ are obtained from their respective docking modules. The pseudo code is provided in Appendix A.2.

Notably, we adopt a partial teacher-forcing (Lamb et al., 2016) strategy during training process. When passing pocket information from the pocket prediction module to the docking modules, a probability factor $p$ is set to control whether the true pocket sites (with probability $p$) or the predicted pocket sites (with probability $1 - p$) are passed to the docking modules. This strategy allows the model to rely partially on ground-truth pocket sites for stability, while progressively learning to depend on its own pocket predictions.

## 4 EXPERIMENTS

### 4.1 EXPERIMENTAL SETTING

**Dataset Construction.** Our experiments are conducted on the widely used public PDBBind v2020 dataset[1], which contains a comprehensive collection of 19,443 protein-ligand crystal complex structures with experimentally measured binding affinities. For each complex, we align the protein component with its corresponding AlphaFold2-predicted structure to obtain the apo protein conformations. To ensure consistency with previous work (Pei et al., 2024; Lu et al., 2024), we employ the same dataset splitting and adhere to similar data preprocessing steps. Specifically, complexes deposited before 2019 are utilized as the training set (12,807 complexes) and validation set (734 complexes), while those recorded after 2019 are designated as test set (303 complexes). Additional data preprocessing details are provided in Appendix B.1.

**Baselines.** We compare our proposed FABFlex against a spectrum of competitors, which are categorized into traditional docking software and deep learning docking methods. Within the traditional software category, we compare against well-established software including Vina (Trott & Olson, 2010), Glide (Friesner et al., 2004) and Gnina (McNutt et al., 2021). For deep learning-based methods, we include TankBind (Lu et al., 2022), FABind (Pei et al., 2024), FABind+ (Gao et al., 2024),

---

[1]http://pdbbind.org.cn/

Table 1: Ligand performance comparison of blind flexible docking.

| Method | Ligand RMSD | | | | | | | | | | | | Average Runtime (s) |
|---|---|---|---|---|---|---|---|---|---|---|---|---|---|
| | On All Cases | | | | | | On Unseen Protein Receptors | | | | | | |
| | Percentiles ↓ | | | | % Below ↑ | | Percentiles ↓ | | | | % Below ↑ | | |
| | 25% | 50% | 75% | Mean | < 2Å | < 5Å | 25% | 50% | 75% | Mean | < 2Å | < 5Å | |
| *Traditional Docking Software* | | | | | | | | | | | | | |
| Vina | 4.79 | 7.14 | 9.21 | 7.14 | 6.67 | 27.33 | 5.27 | 7.06 | 8.84 | 7.15 | 6.25 | 23.21 | 205* |
| Glide | 2.84 | 5.77 | 8.04 | 5.81 | 14.66 | 40.60 | 2.38 | 5.01 | **7.17** | **5.21** | 21.36 | 49.51 | 1405* |
| Gnina | 2.58 | 5.17 | 8.42 | 5.76 | 19.32 | 48.47 | 2.03 | 4.96 | 7.35 | 5.33 | 24.55 | 50.91 | 146 |
| *Deep Learning-based Rigid Docking Methods* | | | | | | | | | | | | | |
| TankBind | 2.82 | 4.53 | 7.79 | 7.79 | 8.91 | 54.46 | 2.88 | 4.45 | 7.53 | 7.60 | 4.39 | 58.77 | 0.87 |
| FABind | 2.19 | 3.73 | 8.39 | 6.63 | 22.11 | 60.73 | 2.73 | 4.83 | 9.35 | 7.15 | 8.77 | 50.88 | 0.12 |
| FABind+ | 1.58 | **2.79** | 6.69 | 5.63 | 35.64 | **66.01** | 1.93 | **3.13** | 8.59 | 6.76 | 27.19 | 57.89 | 0.16 |
| DiffDock | 1.82 | 3.92 | 6.83 | 6.07 | 29.04 | 60.73 | 1.97 | 4.82 | 8.03 | 7.41 | 26.32 | 51.75 | 82.83 |
| DiffDock-L | 1.55 | 3.22 | 6.86 | 5.99 | 36.75 | 62.58 | 1.86 | 3.16 | 9.09 | 7.14 | 29.82 | **61.40** | 58.72 |
| *Deep Learning-based Flexible Docking Methods* | | | | | | | | | | | | | |
| DynamicBind | 1.57 | 3.16 | 7.14 | 6.19 | 33.00 | 64.69 | 2.23 | 4.02 | 10.23 | 8.27 | 20.18 | 54.39 | 102.12 |
| FABFlex | **1.40** | 2.96 | **6.16** | **5.44** | **40.59** | **68.32** | **1.81** | 3.51 | 8.03 | 7.17 | **32.46** | 59.65 | 0.49 |

**Notes:** The best results are highlighted in bold, and the second best results are underlined. The average runtime for each method is presented in seconds. The asterisk (*) indicates that the method is executed on the CPU. The left part of the table compares ligand RMSD on all test cases, while the right part provides a more rigorous comparison for those cases involving protein receptors that were unseen during training process.

DiffDock (Corso et al., 2023), DiffDock-L (Corso et al., 2024) and DynamicBind (Lu et al., 2024). Among these, TankBind, FABind, FABind+, DiffDock and DiffDock-L maintain the rigid protein assumption, while DynamicBind is a recently published flexible docking method. From another perspectives, TankBind, FABind and FABind+ are rooted in regression-based approaches, whereas DiffDock, DiffDock-L and DynamicBind employ a sampling-based strategy with diffusion models to refine the conformations of proteins and ligands. Further details are provided in Appendix B.2.

**Evaluation Metrics.** Our primary evaluation metric is the root-mean-square deviation (RMSD) of Cartesian coordinates, which assesses the accuracy of predicted versus ground-truth structures for both ligands (Ligand RMSD) and binding pockets (Pocket RMSD). These metrics reflect the capability of docking model in accurately predicting ligand structures at atom level and pocket structures at residue level. For Ligand RMSD, we report the percentile values (25%, 50%, 75% and mean) and percentages of RMSD below the threshold ($< 2$Å and $< 5$Å).

**Implementation Settings.** The initial apo ligand structures are generated using the Experimental-Torsion Knowledge Distance Geometry (ETKDG) algorithm (Riniker & Landrum, 2015) in RD-Kit (Landrum et al., 2013) from the molecules' SMILES sequences. The initial apo protein structures are obtained through AlphaFold2 (Jumper et al., 2021) predictions based on the proteins' amino acid sequences. The more details of hyperparameter setting are provided in Appendix B.3.

## 4.2 Ligand Performance of Blind Flexible Docking

Table 1 summarizes the comparison of ligand performance across various docking methods. The left part of the table performs the comparison on the all test cases. It can be observed that FABFlex consistently outperforms both traditional docking software and contemporary deep learning-based methods almost across all metrics. Typically, a prediction of a ligand's structure is considered successful if its predicted structure is within a RMSD of 2Å from the true holo ligand structure (Lu et al., 2024). Thus, the ligand RMSD $< 2$Å is a critical metric to evaluate the capacity of molecular docking methods. FABFlex excels in this metric, achieving the ligand RMSD $< 2$Å at 40.59%, which has generated a significant margin compared to the second-best competitor, showcasing FABFlex's superior ability in predicting accurate binding structures of ligands.

The right part of Table 1 performs a more rigorous assessment of docking methods using 114 ligand-protein complexes that involve protein receptors that were not seen during the training process. This assessment is crucial for evaluating the generalization capability of each method. Although FABFlex does not achieve the best results in every metric, it excels significantly in the critical measure of ligand RMSD $< 2$Å. Notably, FABFlex reaches 32.46%, markedly outperforming all competitors, which fall below 30%. This performance underscores FABFlex's ability to generalize effectively to new and unseen proteins. Additional analysis of ligand performance is provided in Appendix C.1.

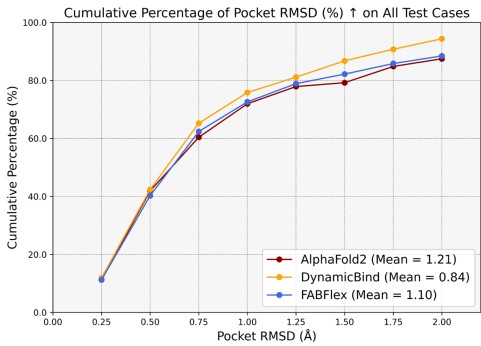 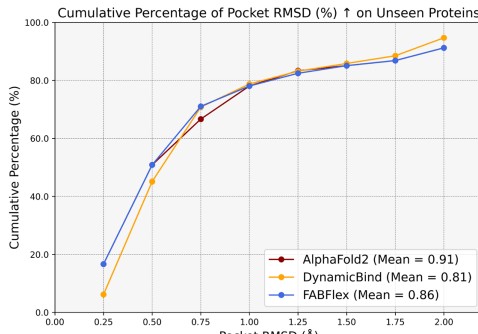

Figure 3: Pocket performance comparison of blind flexible docking. The left figure shows the cumulative percentage of pocket RMSD on all test cases, while the right figure evaluates on those cases with protein receptors that were unseen during training process.

## 4.3 Pocket Performance of Blind Flexible Docking

Fig. 3 illustrates the cumulative distribution of pocket RMSD performance. In the left figure, which evaluates all test complexes, we observe that although FABFlex does not outperform DynamicBind, it still has a positive effect on refining pocket poses from the initial AlphaFold2 structures. The right figure focuses on complexes with unseen protein receptors. It can be observed that for pocket RMSD below 0.75Å, the DynamicBind distribution curve almost always stays at the lowest position, while FABFlex maintains results that are not worse than those of AlphaFold2. This suggests that for these unseen protein receptors, FABFlex demonstrates better robustness compared to DynamicBind, as it does not degrade the protein structures. These outcomes are not surprising, as DynamicBind employs a diffusion model to adjust the conformation of the entire protein, rather than just the pocket region. While this approach reduces the degrees of freedom and enhances the accuracy, it may limit its ability to generalize to new proteins, and face computational efficiency challenges.

## 4.4 Inference Efficiency

High efficiency facilitates the widespread adoption of a method in real-world applications. In the last column on the right of Table 1, we showcase a comparison of the average inference time for each ligand-protein pair. Traditional docking software such as Vina, Glide, and Gnina exhibit notably longer inference times. Among regression-based methods, TankBind, FABind, and FABind+ demonstrate considerably faster speed than sampling-based approaches such as DiffDock and DynamicBind. Notably, FABFlex achieves an inference speed of only 0.49 seconds, which is approximately 208 times faster than DynamicBind, a recently developed method for flexible docking that averages 102.12 seconds. This efficiency gain is attributed to the design philosophy of FABFlex that directly predicts the bound structures without requiring multiple rounds of sampling or extra external pocket detection tools. Additionally, FABFlex specifically focuses on the conformational changes in the core pocket region, rather than the entire protein. As the pocket constitutes a small part of the protein, this design effectively reduces the computational burden.

Table 2: Performance of pocket prediction.

| Method | Given Apo Proteins | | | | Given Holo Proteins | | | |
| | CLS ACC % ↑ | Pocket Center (Å) | | | CLS ACC % ↑ | Pocket Center (Å) | | |
| | | MAE ↓ | RMSE ↓ | EucDist ↓ | | MAE ↓ | RMSE ↓ | EucDist ↓ |
|---|---|---|---|---|---|---|---|---|
| P2Rank | - | 4.04 | 5.69 | 7.85 | - | 4.11 | 6.17 | 8.05 |
| FABind | 73.32 | **3.06** | 5.30 | **6.18** | 73.54 | **3.08** | 5.35 | **6.18** |
| FABind+ | **87.18** | 3.49 | 5.09 | 6.94 | **87.52** | 3.50 | 5.13 | 6.69 |
| FABFlex | 87.08 | 3.29 | **4.83** | 6.59 | 87.06 | 3.31 | **4.89** | 6.63 |

**Notes:** "-" means that P2Rank is an external pocket detection tool without pocket residue classification.

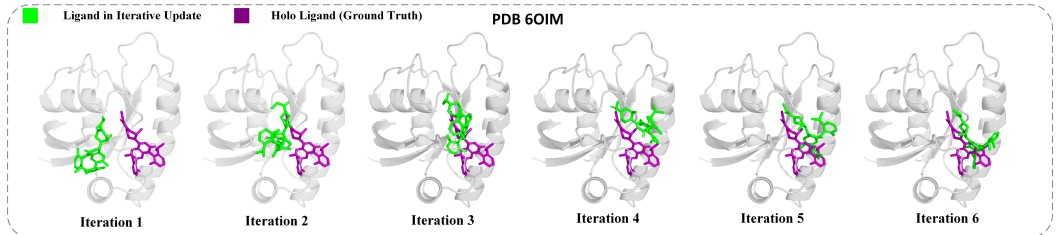

Figure 4: Case PDB 6OIM to intuitively present the process of iterative update.

## 4.5 POCKET PREDICTION ANALYSIS

Table 2 illustrates the performance of FABFlex's pocket prediction module in comparison with to existing P2Rank, FABind and FABind+. Among them, P2Rank (Krivák & Hoksza, 2018) is an open-source tool widely used in existing docking methods for pre-determining potential binding pocket sites (Lu et al., 2022; Zhang et al., 2023). To intuitively analyze the effectiveness of pocket prediction across different methods, we evaluate the quality of pocket prediction from two perspectives: residue classification and pocket center position. For classification, we report the accuracy (CLS ACC). For the pocket center, we employ the mean-absolute-error (MAE), the root-mean-square-error (RMSE), and the Euclidean distance (EucDist), each calculated between predicted pocket center and native pocket center to quantify the accuracy of pocket localization. From Table 2, we can observe that regardless of apo or holo protein, FABFlex predicts pocket sites that are comparable to FABind and FABind+, and consistently outperform P2Rank. These results indicate the effectiveness of FABFlex in identifying pocket residues, even when confronting apo proteins. Additionally, integrating pocket prediction into the flexible docking process appears to be more advantageous than relying on an external pocket detection tool. Notably, FABFlex achieves the lowest RMSE at 4.83 and 4.89, and the lower RMSE suggests FABFlex's stability in error control, with fewer cases of extreme errors. Moreover, we provide intuitive visualization of two pocket prediction cases in Appendix C.12.

Table 3: Experimental results of ablation studies.

| Method | Ligand RMSD (Å) | | | Pocket RMSD (Å) | | Pocket Center (Å) | | |
|---|---|---|---|---|---|---|---|---|
| | Mean ↓ | Median ↓ | <2Å(%) ↑ | Mean ↓ | Median ↓ | MAE ↓ | RMSE ↓ | EucDist ↓ |
| FABFlex | 5.44 | 2.96 | 40.59 | 1.10 | 0.63 | 3.29 | 4.83 | 6.59 |
| One Docking Module | 5.73 | 3.91 | 27.06 | 1.59 | 1.08 | 3.46 | 4.83 | 6.90 |
| w/o Iterative Update | 6.14 | 3.78 | 19.80 | 1.11 | 0.63 | 3.33 | 4.88 | 6.67 |
| Iterative Internally | 5.35 | 2.94 | 35.31 | 1.42 | 0.88 | 3.36 | 4.86 | 6.73 |
| Using P2Rank | 7.28 | 3.07 | 34.22 | 1.15 | 0.68 | 4.04 | 5.69 | 7.85 |

## 4.6 ABLATION STUDY

We conduct a series of ablation studies to investigate different factors affecting model performance, including the following: (1) Using a single docking module to predict both the holo structures of the ligand and pocket degrades performance for both, indicating that decomposing flexible docking into two subtasks helps reduce the complexity each module needs to handle. (2) Removing iterative update mechanism significantly impairs the ligand performance (with ligand RMSD < 2Å from 40.59% to 19.80%), indicating the critical role of iterative update in ligand coordinate refinement. (3) Applying the iterative update mechanism only internally within the ligand and pocket docking modules negatively impacts both ligand RMSD and pocket RMSD, underscoring the importance of prediction exchange to connect the two docking modules. (4) Replacing the pocket sites predicted by the pocket prediction module with those predicted by P2Rank reduces overall performance, suggesting that integrating pocket prediction within the docking process may be more effective than relying on an external pocket detection tool.

## 4.7 ANALYSIS OF ITERATIVE UPDATE

We take a case of PDB 6OIM as an example to visualize the structural refinements in iterative update process to intuitively analyze its function in FABFlex. As shown in Fig. 4, we observe

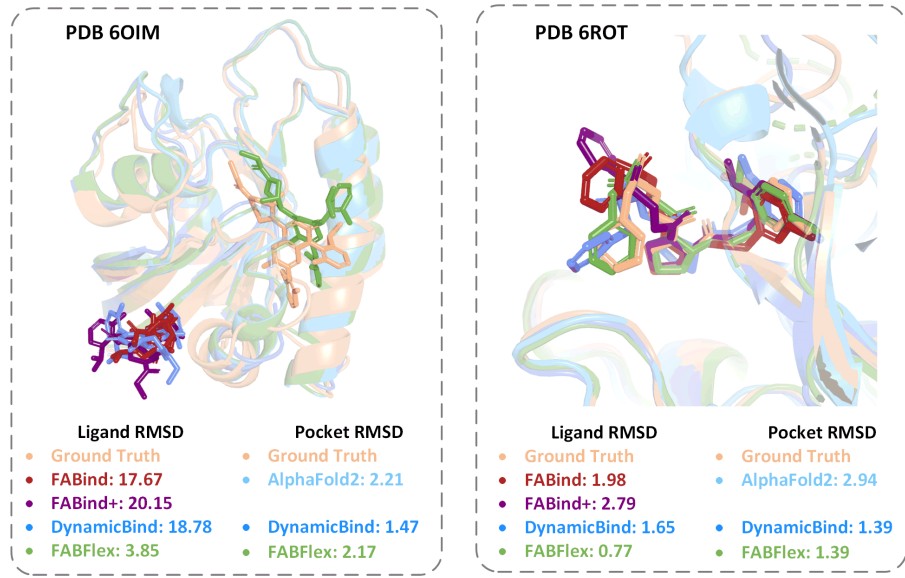

Figure 5: Two case studies of PDB 6OIM (left) and PDB 6ORT (right).

that the ligand gradually approaches the true position of holo ligand from iteration 1 to iteration 6. This resembles a simulation of the ligand docking process, where the molecule is attracted to and interacts with the binding pocket sites. This observation suggests the effectiveness and reasonability of the iterative update mechanism adopted in FABFlex. Additionally, we notice that as iterations increase, the structural changes become smaller, indicating that an appropriate number of iterations is sufficient. More iterations appear unnecessary and may even increase the computational burden without significant benefit. More case studies are provided in Appendix C.7 for further analysis.

## 4.8 CASE STUDY

Fig. 5 visualizes two cases of PDB 6OIM and PDB 6ROT, to demonstrate the effectiveness of our FABFlex in blind flexible docking. We have following observations:

**FABFlex can pinpoint the binding pocket site.** In the case of PDB 6OIM, existing methods such as FABind, FABind+ and DynamicBind wrongly determine the binding sites. In contrast, FABFlex successfully locates the correct binding pockets, achieving significantly lower ligand RMSD of 3.85Å. This underscores FABFlex's ability to accurately pinpoint binding pockets, even in such difficult case where other methods fall short.

**FABFlex excels in ligand structure prediction.** In the case of PDB 6ROT, while all methods successfully identify the correct binding pocket, the ligand structures predicted by FABFlex are significantly closer to the ground truth compared to other methods, with ligand RMSD of merely 0.77Å. This low ligand RMSD means that the ligand predicted by FABFlex is nearly identical to the actual holo ligand. This visualization suggests FABFlex's proficiency in predicting ligand structure. Additional case studies are provided in Appendix C.11.

## 5 CONCLUSION

A high-efficiency molecular docking method is an effective tool in drug discovery, as it enables the fast assessment and screening of millions or even billions of potential molecule-protein interactions within a limited time, discovering promising drug candidates early in the development process. This work proposes FABFlex, a regression-based, end-to-end neural model tailored for real-world blind flexible docking scenarios. FABFlex offers a solution, different from existing generative model-based sampling approaches, to achieve both fast computational speed and accurate docking performance. Looking ahead, we aim to explore fast and accurate multi-site docking, a task of great significance and heightened complexity, as it allows for the investigation of intricate interactions across multiple binding sites.

ACKNOWLEDGMENTS

ZZZ and BH were supported by RGC Young Collaborative Research Grant No. C2005-24Y, NSFC General Program No. 62376235, Guangdong Basic and Applied Basic Research Foundation Nos. 2022A1515011652 and 2024A1515012399, MSRA StarTrack Scholars Program, HKBU Faculty Niche Research Areas No. RC-FNRA-IG/22-23/SCI/04, and HKBU CSD Departmental Incentive Scheme. JCY was supported by National Natural Science Foundation of China (No. 62306178), 111 plan (No. BP0719010).

## 6 ETHICS STATEMENT

This paper proposes a model aimed at enhancing both computational efficiency and accuracy in blind flexible docking. All data used for model training and evaluation were obtained from publicly available molecular docking benchmark datasets. The advancement of molecular docking technologies has both positive and potential negative implications. On the positive side, our method can contribute to advancements in drug discovery. However, there is also the risk of misuse or malicious use, as molecular docking models could be applied to explore harmful or non-therapeutic compounds. To mitigate this risk, we are releasing our code with the explicit intent of supporting ethical and medically approved drug discovery efforts, in full compliance with relevant legal standards and institutional guidelines. This paper adheres to strict research integrity protocols and responsible dataset usage, ensuring that our contributions support the broader objectives of ethical scientific progress.

## 7 REPRODUCIBILITY STATEMENT

The experimental settings for our model are described in detail in Section 4.1. The experiments are all conducted on the widely used public molecular docking benchmark PDBBind v2020 dataset. We provide the link to our source codes to ensure the reproducibility of our experimental results: https://github.com/tmlr-group/FABFlex

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

# A  METHODOLOGY DETAILS

## A.1  DETAILS OF TRAINING LOSSES

The blind flexible docking is modelled as a supervised learning task. The ground-truth encompasses residues that form binding pocket sites, along with the actual coordinates of both holo ligand and holo pocket. The training loss implementations draw extensively from the methodologies outlined in FABind (Pei et al., 2024) and FABind+ (Gao et al., 2024). Corresponding to the statement in Section 3.2.4, we provide detailed descriptions of each training loss to improve the reproducibility of our study:

Given that only a small fraction of the residues in a protein belong to the pocket, the pocket residue detection is treated as an imbalanced binary classification task using binary-classification cross entropy loss (BCELoss)[2], aiming at identifying which residues belongs to the pocket. The formulation of this loss is given by:

$$\mathcal{L}_{\text{pocket\_cls}} = \frac{1}{N} \sum_{i=1}^{N} \frac{p_i}{q_i} \{ - \sum_{j=1}^{p_i} [y_j \log(\hat{y}_j) + (1 - y_j) \log(1 - \hat{y}_j)] \}, \tag{6}$$

where $N$ is the total number of training complexes, $p_i$ the number of residues in the $i$-th protein, $q_i$ the count of residues that form the pocket in the $i$-th protein. The weighting factor $p_i/q_i$ adjusts the emphasis on proteins with fewer pocket-forming residues, ensuring their adequate importance in the training process. Additionally, the pocket center loss, denoted as $\mathcal{L}_{\text{pocket\_center}}$, adopts a Huber loss[3] to supervise the position of the predicted pocket, formulated as follows:

$$\mathcal{L}_{\text{pocket\_center}} = \frac{1}{N} \sum_{i=1}^{N} \text{HuberLoss}(center_i, \widehat{center}_i), \tag{7}$$

where $center_i$ is the actual centroid coordinate of pocket in the $i$-th protein, and $\widehat{center}_i$ is the predicted centroid coordinate, which is calculated using a weighted average of all residue coordinates in the $i$-th protein, with the weights derived from the output probabilities of the Gumbel-Softmax distribution. After that, the residue classification loss $\mathcal{L}_{\text{pocket\_cls}}$ and the pocket center loss $\mathcal{L}_{\text{pocket\_center}}$, are combined to formulate the overall pocket prediction loss $\mathcal{L}_{\text{pocket\_pred}}$ as follows:

$$\mathcal{L}_{\text{pocket\_pred}} = \alpha_1^{\text{cls}} \mathcal{L}_{\text{pocket\_cls}} + \alpha_1^{\text{center}} \mathcal{L}_{\text{pocket\_center}}. \tag{8}$$

The ligand coordinate loss and the pocket coordinate loss both utilize the Huber loss to regress the predicted structures towards their respective real holo structures:

$$\mathcal{L}_{\text{ligand\_coord}} = \frac{1}{N} \sum_{i=1}^{N} \text{HuberLoss}(\tilde{\mathbf{x}}_i^l, \hat{\mathbf{x}}_i^l), \quad \mathcal{L}_{\text{pocket\_coord}} = \frac{1}{N} \sum_{i=1}^{N} \text{HuberLoss}(\tilde{\mathbf{x}}_i^p, \hat{\mathbf{x}}_i^p), \tag{9}$$

where $\hat{\mathbf{x}}_i^l$ ($\hat{\mathbf{x}}_i^p$) and $\tilde{\mathbf{x}}_i^l$ ($\tilde{\mathbf{x}}_i^p$) are the predicted ligand (pocket) coordinates and actual holo ligand (pocket) coordinates respectively, for the $i$-th complex.

The distance map loss serves as an auxiliary objective, employing mean-squared-error loss (MSELoss)[4] to supervise the relative positions between ligand atoms and pocket residues, formally expressed as follows:

$$\mathcal{L}_{\text{dis\_map}} = \frac{1}{N} \sum_{i=1}^{N} \text{MSELoss}(\tilde{\mathbf{D}}_i, \hat{\mathbf{D}}_i), \tag{10}$$

where $\tilde{\mathbf{D}}_i$ and $\hat{\mathbf{D}}_i$ represent the actual and predicted distance maps respectively. Each element $\tilde{D}_i^{jk}$ in $\tilde{\mathbf{D}}_i$ is the Euclidean distance between the $j$-th atom in the ligand and the $k$-th residue in the pocket. Similarly, $\hat{D}_i^{jk}$ in $\hat{\mathbf{D}}_i$ corresponds to the predicted pairwise distances.

---

[2]https://pytorch.org/docs/stable/generated/torch.nn.BCEWithLogitsLoss.html#bcewithlogitsloss

[3]https://pytorch.org/docs/stable/generated/torch.nn.HuberLoss.html#huberloss

[4]https://pytorch.org/docs/stable/generated/torch.nn.MSELoss.html#mseloss

---

**Algorithm 1** Pseudo code of FABFlex's inference.

---

1: **Inputs:**
2:     3D coordinates of apo ligands $\{\mathbf{x}_i \in \mathbb{R}^3\}_{1 \leq i \leq n^l}$ and apo proteins $\{\mathbf{x}_j \in \mathbb{R}^3\}_{1 \leq j \leq n^p}$
3:     Ligand atom features $\{\mathbf{h}_i \in \mathbb{R}^{d^l}\}_{1 \leq i \leq n^l}$ and protein residue features $\{\mathbf{h}_j \in \mathbb{R}^{d^p}\}_{1 \leq j \leq n^p}$
4: **Outputs:**
5:     Predicted coordinates of holo ligand $\{\hat{\mathbf{x}}_i \in \mathbb{R}^3\}_{1 \leq i \leq n^l}$ and holo pockets $\{\hat{\mathbf{x}}_j \in \mathbb{R}^3\}_{1 \leq j \leq \hat{n}^{p*}}$
6: **Inference:**
7:     $\{\hat{y}_j\}_{1 \leq j \leq n^p} = \mathcal{M}_S(\mathcal{G}, \{\mathbf{h}_i, \mathbf{x}_i\}_{1 \leq i \leq n^l}, \{\mathbf{h}_j, \mathbf{x}_j\}_{1 \leq j \leq n^p})$,    *# predict pocket sites*
8:     $\{v_j\}_{1 \leq j \leq \hat{n}^{p*}} = \{\hat{y}_j \odot v_j\}_{1 \leq j \leq n^p}$,    *# pick out pocket residues with $\hat{y}_j = 1$*
9: **for** iterative update $k : 1 \rightarrow K$ **do**
10:     $\hat{\mathcal{G}}^{(k)*} \xleftarrow{\text{construct}} \{\mathcal{G}^l, \{\hat{\mathbf{x}}_i^{(k)}\}_{1 \leq i \leq n^l}, \{\hat{\mathbf{x}}_j^{(k)}\}_{1 \leq j \leq \hat{n}^{p*}}\}$,    *# construct ligand-pocket graph*
11:     $\{\hat{\mathbf{x}}_i^{(k+1)}\}_{1 \leq i \leq n^l} = \mathcal{M}_L(\hat{\mathcal{G}}^{(k)*}, \{\mathbf{h}_i, \hat{\mathbf{x}}_i^{(k)}\}_{1 \leq i \leq n^l}, \{\mathbf{h}_j, \hat{\mathbf{x}}_j^{(k)}\}_{1 \leq j \leq \hat{n}^{p*}})$,    *# predict ligand*
12:     $\{\hat{\mathbf{x}}_j^{(k+1)}\}_{1 \leq j \leq \hat{n}^{p*}} = \mathcal{M}_P(\hat{\mathcal{G}}^{(k)*}, \{\mathbf{h}_i, \hat{\mathbf{x}}_i^{(k)}\}_{1 \leq i \leq n^l}, \{\mathbf{h}_j, \hat{\mathbf{x}}_j^{(k)}\}_{1 \leq j \leq \hat{n}^{p*}})$,    *# predict pocket*
13: **end for**
14: $\{\hat{\mathbf{x}}_i\}_{1 \leq i \leq n^l} \leftarrow \{\hat{\mathbf{x}}_i^{(K)}\}_{1 \leq i \leq n^l}, \{\hat{\mathbf{x}}_j\}_{1 \leq j \leq \hat{n}^{p*}} \leftarrow \{\hat{\mathbf{x}}_j^{(K)}\}_{1 \leq j \leq \hat{n}^{p*}}$,    *# final predicted structures*
15: **Return:** Predicted coordinates of holo ligand $\{\hat{\mathbf{x}}_i\}_{1 \leq i \leq n^l}$ and holo pocket $\{\hat{\mathbf{x}}_j\}_{1 \leq j \leq \hat{n}^{p*}}$.

---

## A.2 PSEUDO CODE OF FABFLEX

To intuitively elucidate the comprehensive inference processes of FABFlex, we delineate the pseudo code in Algorithm 1. The inference begins with the pocket prediction module, which identifies the binding pocket sites. Subsequently, the ligand and pocket docking modules iteratively refine and predict the holo structures of ligands and pockets.

## B EXPERIMENTAL DETAILS

### B.1 DETAILS OF DATASET PREPROCESSING

PDBBind v2020 is a widely utilized benchmark database in related molecular docking research (Pei et al., 2024; Lu et al., 2024; 2022; Corso et al., 2023). It collected 19,443 experimentally measured protein-ligand complexes along with their 3D structures. Due to the absence of apo protein structures in the dataset, we follow Lu et al. (2024) and employ the well-established AlphaFold2 to predict the apo conformations of these protein structures. Consistent with above mentioned studies, we adopt the same dataset split strategy to enhance comparability. From the dataset, we select 303 test complexes recorded after 2019 and 734 validation complexes recorded before 2019, with the remaining complexes allocated for training. For the training set, complex samples that could not be processed by RDKit (Landrum et al., 2013) or TorchDrug (Zhu et al., 2022) are excluded. Further filtering excludes samples with protein amino acid chains longer than 1500 residues and molecules larger than 150 heavy atoms, resulting in a refined set of 12,807 training complexes. The statistics of the dataset are summarized in Table 4.

Table 4: Dataset statistics of preprocessed PDBBind v2020.

| PDBbind | # Complexes | avg. residues in proteins | max. residues in proteins | avg. heavy atoms in ligands | max. atoms in ligands |
|---------|-------------|---------------------------|---------------------------|-----------------------------|-----------------------|
| Train | 12,807 | 310.59 | 1,290 | 31.30 | 149 |
| Validate | 734 | 312.14 | 1,025 | 32.52 | 177 |
| Test | 303 | 280.92 | 1,098 | 35.77 | 147 |

### B.2 BASELINES

We compare our proposed method with a variety of competitors, including traditional molecular docking software and recent deep learning-based methods:

Table 5: Implementation configuration of FABFlex.

| Configuration | | Pretraining Stage 1 | Pretraining Stage 2 | Joint Training Stage |
|---|---|---|---|---|
| Learning rate | | 5e-5 | 5e-5 | 5e-6 |
| Epoch | | 1000 | 600 | 600 |
| Batch size | | 2 | 4 | 4 |
| Dropout | | 0.1 | 0.1 | 0.1 |
| Teacher-forcing | | Yes | Yes | No ($p = 0.5$) |
| Optimizer | | Adam | Adam | Adam |
| Scheduler | | LinearLR | LinearLR | LinearLR |
| Loss weights | Residue classification $\alpha_1^{\text{cls}}$ | 1.0 | - | 1.0 |
| | Pocket center $\alpha_1^{\text{center}}$ | 0.05 | - | 0.05 |
| | Ligand docking $\alpha_2$ | 1.5 | - | 1.5 |
| | Pocket docking $\alpha_3$ | - | 15.0 | 15.0 |
| | Distance map $\alpha_4$ | 1.0 | 1.0 | 1.0 |

- **Traditional molecular docking software**:
  - **Vina** (Trott & Olson, 2010): AutoDock Vina is among the most widely used open-source docking programs, renowned for its enhanced docking performance. It achieves this through improvements to its scoring function, optimization algorithm and search strategy. Specifically, Vina enhances the X-score (Wang et al., 2002) and employs the Broyden-Fletcher-Goldfarb-Shanno (BFGS) (Nocedal & Wright, 1999) method for local optimization.
  - **Glide** (Friesner et al., 2004): Glide distinguishes itself from other docking software by performing a systematic and comprehensive search of the conformational, orientational, and positional spaces of the docked ligand to a rigid protein receptor. This process involves technologies such as hierarchical filters, ChemScore function (Eldridge et al., 1997), among others.
  - **Gnina** (McNutt et al., 2021): Gina is an advanced molecular docking software that incorporates convolutional neural networks into its scoring function, and leverages Monte Carlo sampling to comprehensively explore the conformational space of ligands.
- **Deep learning-based methods following protein rigidity assumption**:
  - **TankBind** (Lu et al., 2022): TankBind employs the external tool P2Rank (Krivák & Hoksza, 2018) to detect potential binding pocket sites, segmenting the entire protein into functional blocks. Subsequently, TankBind uses a trigonometry-aware graph neural network to model protein-ligand interactions, predicting the distance matrix and optimizing ligand structures.
  - **FABind** (Pei et al., 2024): FABind adopts an end-to-end framework that streamlines the modeling of protein-ligand interactions by integrating binding pocket predictions with docking tasks. This approach accelerates the docking computation process by eliminating the need for external pocket detection tools.
  - **FABind+** (Gao et al., 2024): FABind+ is an enhanced version of FABind, incorporating additional improvements such as dynamic pocket radius adjustments for each complex and permutation invariance loss to supervise ligand structures. These enhancements boost the performance of predicted ligand structures.
  - **DiffDock** (Corso et al., 2023): DiffDock utilizes diffusion models (Yang et al., 2023) to manage the conformational refinements of ligands in molecular docking. This approach conceptualizes docking as a generative modeling task by mapping ligand poses onto a non-Euclidean manifold, effectively reducing the degrees of freedom for structural refinements. Additionally, DiffDock trains a confidence model to estimate the poses for multiple samplings.
  - **DiffDock-L** (Corso et al., 2024): DiffDock-L is an enhanced version of DiffDock (Corso et al., 2023) that improves docking model performance by scaling data and model size and incorporating synthetic data to boost generalization capacity.
- **Recent deep learning-based flexible docking method**:
  - **DynamicBind** (Lu et al., 2024): DynamicBind is a recent flexible docking method breaking down the rigid protein assumption. It leverages equivariant geometric diffusion networks to manipulate the structure changes in both ligands and proteins, recovering ligand-specific poses from their apo stages.

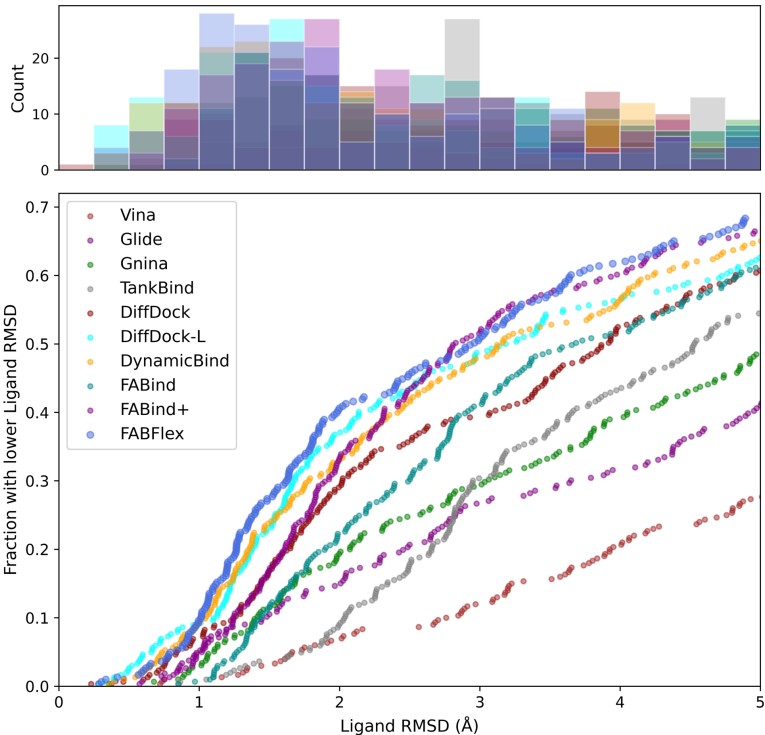

Figure 6: The cumulative distribution of ligand RMSD.

### B.3 DETAILS OF IMPLEMENTATION SETTING

**Model Configuration.** The dimension $d^l$ of initial features extracted via TorchDrug (Zhu et al., 2022) for ligand atoms is set to 56, and the dimension $d^p$ of ESM-2 (Lin et al., 2022) features for amino acid is 1280. The number of FABind-layers is configured as $\{1, 5, 5\}$ for pocket prediction module, ligand docking module and pocket docking module, respectively. The number of hidden size is set to $\{128, 512, 512\}$ for the same modules in the corresponding order. The pocket radius of 20Å is used to delineate the binding pocket sites for each ligand-protein complex.

**Training details.** Simultaneously optimizing all three subtasks of blind flexible docking from scratch is a challenging task. We introduce a two-stage pretraining process to warm up the model. In the first stage, the model is trained under rigid docking conditions, using a (holo protein, apo ligand) pair to predict the holo ligand structure. In the second stage, the model is trained using an (apo protein, holo ligand) pair to predict the holo pocket structure. After that, the model undergoes joint training to predict both the holo ligand and holo pocket from (apo protein, apo ligand) inputs. This progressive approach allows the model to learn from simpler tasks to more complex ones, facilitating the solution of blind flexible docking.

**Training Configuration.** Table 5 summarizes the training configurations in this study. The experiments are conducted using the Pytorch framework[5]. The model is trained on eight NVIDIA RTX 4090 GPUs. The pretraining stages 1 and 2 reduce task difficulty by fixing either the small molecule or the protein as the holo structure while keeping the other as the apo structure, to warm up the model. The approximate training durations for pretraining stages 1 and 2, as well as the joint training stage, are $\{10, 5, 5\}$ days, respectively. The configurations vary slightly across different stages, mainly in terms of learning rate, and the application of teacher forcing. During the pretraining stages, and teacher forcing are utilized to kick-start the model effectively. As the progress to the joint training stage, the learning rate is reduced to better adapt the model to flexible docking.

---

[5]https://pytorch.org/

Table 6: Analysis of the number of samplings.

| Methods | Ligand RMSD (Å) | | | Pocket RMSD (Å) | | Average Runtime (s) |
|---|---|---|---|---|---|---|
| | Mean ↓ | Median ↓ | <2Å(%) ↑ | Mean ↓ | Median ↓ | |
| DynamicBind(1) | 6.26 | 3.45 | 27.15 | 0.84 | 0.59 | 22.04 |
| DynamicBind(10) | 6.21 | 3.41 | 27.48 | 0.84 | 0.57 | 47.22 |
| DynamicBind(40) | 6.19 | 3.16 | 33.00 | 0.84 | 0.58 | 102.12 |
| FABFlex | 5.44 | 2.96 | 40.59 | 1.10 | 0.63 | 0.49 |

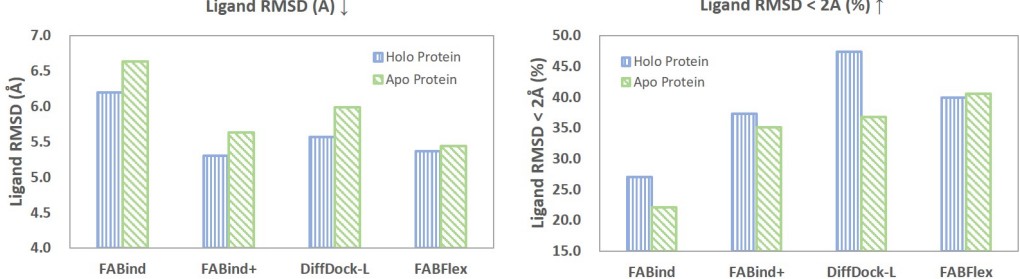

Figure 7: Performance from holo proteins to apo proteins. Left: ligand RMSD. Right: ligand RMSD < 2Å.

## C  ADDITIONAL EXPERIMENTAL RESULTS

### C.1  DISTRIBUTION OF LIGAND PERFORMANCE

To showcase the ligand performance comprehensively, Fig. 6 provides the cumulative distribution of ligand RMSD. It can be observed that FABFlex occupies the topmost position in the distribution curve in the majority of cases. Especially for the cases of ligand RMSD < 2Å, which are crucial indicators for evaluating a molecular docking method, the superiority of FABFlex becomes even more distinct. This findings corroborates previous results in Table 1, further highlighting the advantages of FABFlex in ligand performance.

### C.2  ANALYSIS OF NUMBER OF SAMPLINGS

In this section, we specifically vary the number of samplings in {1, 10, 40} of DynamicBind for further analysis. Table 6 illustrates the experimental results. It can be observed that as the number of samplings decreases, there is a notable performance degradation for DynamicBind, for example, the percentage of ligand RMSD less than 2Å drops from 33.00% with 40 samplings to 27.15% with one samplings. Besides, even DynamicBind(1) only sample once, its runtime is still much slower than that of regression-based models. These observations reflect the inherent trade-off between performance and efficiency in these diffusion-based methods like DynamicBind, where the runtime grows with the number of samplings.

### C.3  PERFORMANCE FROM RIGID TO FLEXIBLE

In this section, we conduct experiments to evaluate the performance differences between rigid protein docking and flexible docking. Fig. 7 illustrates comparative performances when holo and apo proteins are employed as inputs in molecular docking experiments. We observe a notable decrease in performance when transitioning from holo to apo structures across all rigid docking methods including FABind, FABind+ and DiffDock-L. Conversely, FABFlex consistently maintains its performance, whether measuring ligand RMSD or the percentage of ligand RMSD < 2Å. These results indicate that FABFlex's superior ligand performance over the FABind series can be attributed to its self-adaptive capability to accommodate protein conformational changes. This capacity allows FABFlex to effectively handle molecular docking under both rigid protein assumption and flexible docking condition.

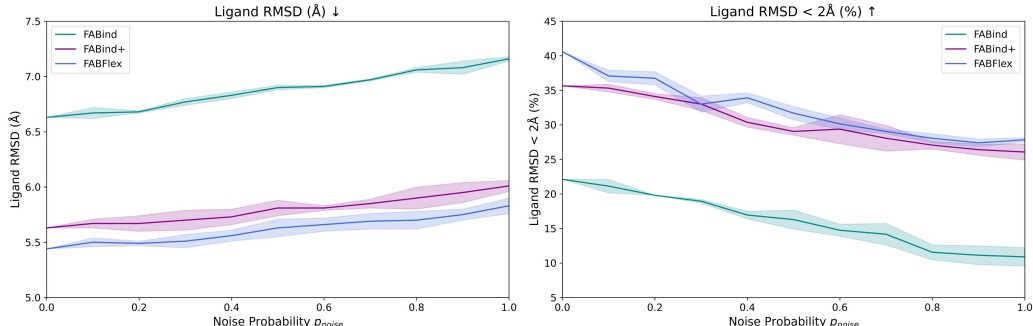

Figure 8: Results of adding noise. Left: ligand RMSD. Right: ligand RMSD < 2Å.

Table 7: Impact of the number of iterative updates.

| Iteration | Ligand RMSD (Å) | | | Pocket RMSD (Å) | | Pocket Center (Å) | | | Average Runtime (s) |
|---|---|---|---|---|---|---|---|---|---|
| | Mean ↓ | Median ↓ | <2Å(%) ↑ | Mean ↓ | Median ↓ | MAE ↓ | RMSE ↓ | EucDist ↓ | |
| # ITER=1 | 6.11 | 3.69 | 21.45 | 1.12 | 0.64 | 3.33 | 4.90 | 6.69 | 0.11 |
| # ITER=2 | 5.50 | 3.01 | 36.96 | 1.13 | 0.65 | 3.31 | 4.86 | 6.63 | 0.18 |
| # ITER=4 | 5.41 | 2.91 | 39.27 | 1.14 | 0.64 | 3.31 | 4.88 | 6.64 | 0.33 |
| # ITER=6 | 5.44 | 2.96 | 40.59 | 1.10 | 0.63 | 3.29 | 4.83 | 6.59 | 0.49 |
| # ITER=8 | 5.33 | 2.87 | 38.94 | 1.10 | 0.63 | 3.28 | 4.79 | 6.58 | 0.60 |

## C.4 ROBUSTNESS EVALUATION

To further evaluate the robustness and corroborate the discussion in Appendix C.3, we conduct the experiments that introduce perturbations into the input protein conformations. Specifically, we incrementally introduce standard Gaussian noise (mean = 0, standard deviation = 1) to the coordinates within the binding pocket. The noise is added based on a probability parameter $p_{noise}$ which varies from 0.1 to 1.0 with interval 0.1, that is, meaning that each coordinate has a probability $p_{noise}$ of having noise added. To enhance the reliability of the results, each experiment is repeated with three different random seeds, and the mean results along with their standard deviations are reported. The results are illustrated in Fig. 8. As expected, the performance of all methods inevitably deteriorates as the noise increases, as observed in the figure. However, the curve corresponding to FABFlex consistently demonstrates the better performance. This suggests that, despite the lack of specific robustness-focused design in FABFlex, it still exhibits a certain degree of robustness, capable of handling a certain level of perturbation in the input protein structures.

## C.5 HYPERPARAMETER ANALYSIS

We conduct experiments to analyze the impact of varying the number of iterative updates. Table 7 summarizes the results. We observe that as the iterations increase, the ligand RMSD metrics improve and tend to stabilize, and the average inference times also increase. In contrast, the improvement in the metrics of pocket RMSD is minimal, and the metrics for pocket center show a slight improvement. These findings underscore the impact of the number of iterative updates on refineing ligand structures, as an appropriate number of iterations can greatly enhance the structures of ligands.

Table 8: Assessment of PoseBuster test suites.

| Methods | Ligand RMSD < 2Å (%) | PB-valid & Ligand RMSD < 2Å (%) |
|---|---|---|
| FABind | 22.52 | 1.32 |
| FABind+ | 34.11 | 8.94 |
| DiffDock-L | 37.42 | 16.89 |
| DynamicBind | 30.79 | 14.57 |
| FABFlex | 39.40 | 13.91 |

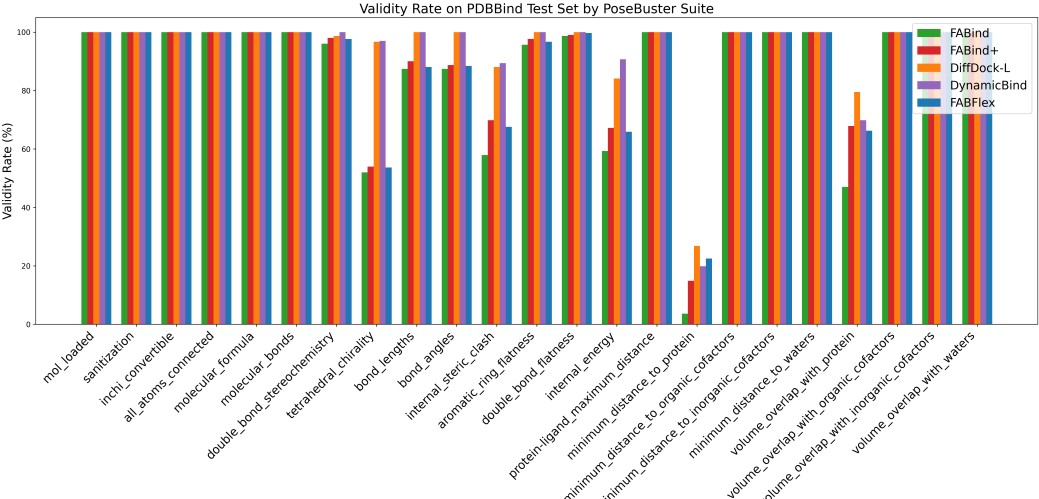

Figure 9: Assessment of 20 aspects by PoseBuster test suites.

## C.6 ASSESSMENT OF POSEBUSTER TEST SUITES

To assess the physical validity of generated ligand structures, we use the PoseBuster (Buttenschoen et al., 2024) test suites performing on the flexible PDBBind to evaluate the ligand structures predicted by various docking models. Table 8 summarizes the experimental results. Following Pose-Buster, we call that molecule poses which pass all tests in PoseBusters are "PB-valid". It can be observed that FABFlex achieves the highest RMSD < 2Å percentage of 39.40% among all competitors, demonstrating its superior accuracy in predicting docking poses close to the ground truth holo ligands. However, the percentage of "PB-valid & RMSD < 2Å" of FABind, FABind+, and FABFlex is lower than those of DiffDock-L and DynamicBind, reflecting the strength of diffusion-based models in generating physically valid molecular poses. This may be attributed to that diffusion-based models adjust molecular structures through translational, rotational, and torsional movements, reducing the degrees of freedom. Notably, FABFlex achieves a "PB-valid & RMSD < 2Å" (13.91%) comparable to DynamicBind (14.57%), larger than poor FABind (1.32%) and FABind+ (8.94%), showcasing its balance in pursuing both accuracy and physical validity. Moreover, to analyze the physical validity comprehensively, we showcase the validity rates of all aspects by PoseBuster test suites in Figure 9. It can be observed that a core limitation regarding the physical validity of deep learning-based docking models lies in the "minimum_distance_to_protein", that is, the distance between protein-ligand atom pairs is larger than 0.75 times the sum of the pairs van der Waals radii. All docking methods demonstrate a low validity rate in this aspect. Additionally, the validity of other critical factors, such as "tetrahedral_chirality", "internal_steric_clash", "internal_energy", and "volume_overlap_with_the_protein", also requires further improvement.

## C.7 ADDITIONAL CASES FOR ANALYSIS OF ITERATIVE UPDATE

Fig. 10 provides additional three cases to further support the discussion in Section 4.7. In case PDB 6PGO and PDB 6GZY, we observe a similar iterative update process as seen in PDB 6OIM. The ligands gradually approach the true holo structures from iteration 1 to iteration 6. The case of PDB 6CJJ is slightly different. We observe that the ligand is already positioned close to the holo ligand at iteration 1. From iterations 1 to 6, the ligand undergoes continuous refinement, and by iteration 6, it aligns very closely with the true holo structure. Consequently, these cases suggest that the iterative update process not only rectifies ligand positions when far from the holo ligands, but also refines the ligand conformations that are already near the correct binding position.

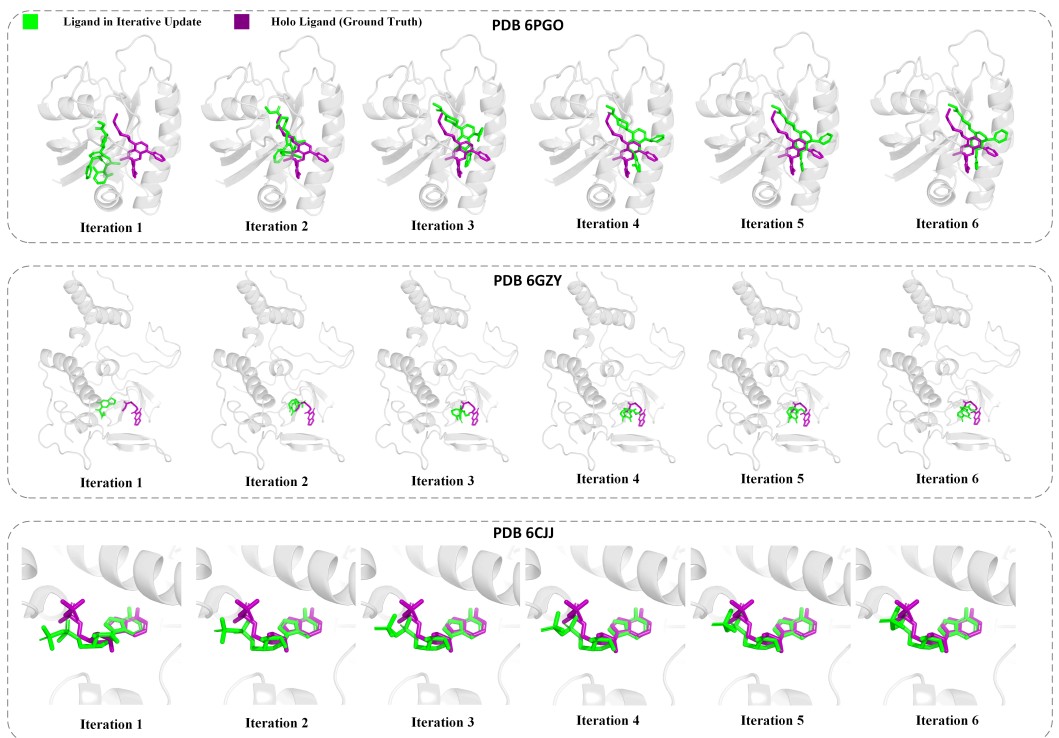

Figure 10: Three more case studies of PDB 6PGO, PDB 6GZY, and PDB 6CJJ, to intuitively show-case the iterative update from iteration 1 to iteration 6.

Table 9: Analysis of clash score.

| Methods | Vina | Glide | Gnina | TankBind | FABind | FABind+ | DiffDock | DynamicBind | FABFlex |
|---|---|---|---|---|---|---|---|---|---|
| clash score ↓ | 0.02 | 0.08 | 0.05 | 0.41 | 0.51 | 0.45 | 0.33 | 0.27 | 0.37 |

## C.8   ANALYSIS OF STERIC CLASH

In this section, we conduct experiments to analyze the steric clash in the predicted structures by the docking methods. Following DynamicBind (Lu et al., 2024), the clash score is defined as the root-mean-square of the van der Waals overlaps for all atom pairs between the ligand and the protein where the interatomic distance is less than 4Å, formulated as follows:

$$\text{clash score} = \sqrt{\frac{\sum_{i=0}^{N} \text{VdW overlop}^2}{N}}, \tag{11}$$

where $N$ is the number of atom pairs with distances considered. A lower clash score indicates fewer or less steric clashes. Table 9 illustrates the class scores across various competitors. It can be observed that traditional docking software achieves significantly lower clash scores, whereas all deep learning-based docking methods exhibit higher clash scores. This reflects a core limitation of deep learning-based docking methods: despite their improved docking accuracy, they are more prone to steric clash issues. Addressing this challenge is a promising direction for future research.

Table 10: Analysis of binding affinity, including IC50 and K value.

| Methods | FABind | FABind+ | DynamicBind | FABFlex |
|---|---|---|---|---|
| IC50 $(\mu M)$ ↓ | 6.32 | 6.24 | 6.27 | 5.92 |
| K Value $(\mu M)$ ↓ | 6.23 | 6.19 | 6.18 | 6.10 |

## C.9 ANALYSIS OF BINDING AFFINITY

In this section, we conduct experiments to evaluate the binding affinity of predicted structures of ligand-protein complexes. Specifically, we employ the MBP (Yan et al., 2024), a structure-based binding affinity prediction model, to evaluate IC50 and K value of the predicted structures as indicators of binding affinity. IC50 represents the concentration of a ligand required to inhibit 50% of the protein's activity, while K Value reflects the dissociation constant, directly measuring the strength of the ligand-protein binding affinity[6]. For both metrics, smaller values indicate stronger binding affinity and better docking quality. Table 10 summarizes the experimental results. It can be observed that FABFlex achieves the lowest IC50 at 5.92 $\mu M$ and K Value at 6.10 $\mu M$ among all competitors. This observation suggests that FABFlex has potential to better predict biologically relevant interactions.

Table 11: Performance comparison in pocket-based flexible docking.

| Methods | Ligand RMSD | | | | Pocket RMSD | | Avg. Runtime (s) |
|---------|-------------|---------------|----------|----------|-------------|----------------|------|
|         | Mean (Å) ↓  | Median (Å) ↓  | < 2Å (%) | < 5Å (%) | Mean (Å) ↓  | Median (Å) ↓   |      |
| FABind              | 4.47 | 3.23 | 23.76 | 67.66 | - | - | 0.10 |
| FABind+             | 4.21 | 2.66 | 36.63 | 69.97 | - | - | 0.14 |
| DiffDock-Pocket(10) | -    | 2.60 | 41.00 | -     | - | - | 17   |
| DiffDock-Pocket(40) | -    | 2.60 | 41.70 | -     | - | - | 61   |
| ReDock(10)          | -    | 2.50 | 39.00 | 74.80 | - | - | 15   |
| ReDock(40)          | -    | 2.40 | 42.90 | 76.40 | - | - | 58   |
| FABFlex             | 3.45 | 2.57 | 42.24 | 75.25 | 0.93 | 0.66 | 0.47 |

## C.10 PERFORMANCE ON POCKET-BASED FLEXIBLE DOCKING

In this section, we conduct experiments to assess the applicability of FABFlex in pocket-based flexible docking, where reliable prior knowledge of the binding pocket sites or sidechains is available. Specifically, the pocket prediction module is bypassed, and the partial teacher-forcing strategy is removed to adapt FABFlex model to pocket-based docking, where we feed the ground truth pocket amino acids and try to predict the holo structures of ligand and pocket. Moreover, we additionally evaluate FABind and FABind+ in pocket-based docking scenario, and we include the results reported in DiffDock-Pocket (Plainer et al., 2023) and ReDock (Huang et al.) for comparison. The experimental results are summarized in Table 11. It can be observed that FABFlex is also effective in pocket-based docking scenario, achieving a Ligand RMSD < 2Å of 42.24%, outperforming FABind, FABind+, DiffDock-Pocket(10), DiffDock-Pocket(40), and ReDock(10), and comparable to ReDock(40) of 42.90%. Furthermore, FABFlex achieves a Pocket RMSD Mean of 0.93 Å, demonstrating its ability to model protein flexibility. On Ligand RMSD < 5Å, FABlex achieves 75.25%, comparable to ReDock(40), surpassing FABind, FABind+ and ReDock(10). Notably, the average runtime of FABFlex is only 0.47 seconds, which is considerably faster than DiffDock-Pocket(40) and ReDock(40) (more than 100x times faster), indicating the strong efficiency of our FABFlex. These results suggest that the regression-based paradigm has the potential and capacity to handle protein flexibility in both blind and pocket-based docking scenarios.

## C.11 ADDITIONAL CASE STUDIES

In this section, we provide six additional cases to further support the discussion in Section 4.8. Fig. 11 visualizes four cases of PDB 6PGO, PDB 6PGP, PDB 6EEB, and PDB 6O5G. FABFlex successfully positions the ligands in the correct pocket regions and achieves the lowest ligand RMSD. Fig. 12 visualizes two cases of PDB 6CYH and PDB 6N8X, in which FABFlex closely approximates the ground-truth ligand structures, with ligand RMSD of 0.94Å and 0.73Å, respectively. Furthermore, FABFlex positively reduces pocket RMSD across all cases. These cases highlight FABFlex's proficiency in accurately identifying pocket sites and predicting ligand structures.

## C.12 CASE STUDIES OF POCKET PREDICTION

In this section, Fig. 13 visualizes two cases of PDB 6QQW and PDB 6HHJ, to intuitively showcase the effectiveness of our FABFlex in binding pocket prediction. It can be observed that for both two

---

[6]https://en.wikipedia.org/wiki/IC50

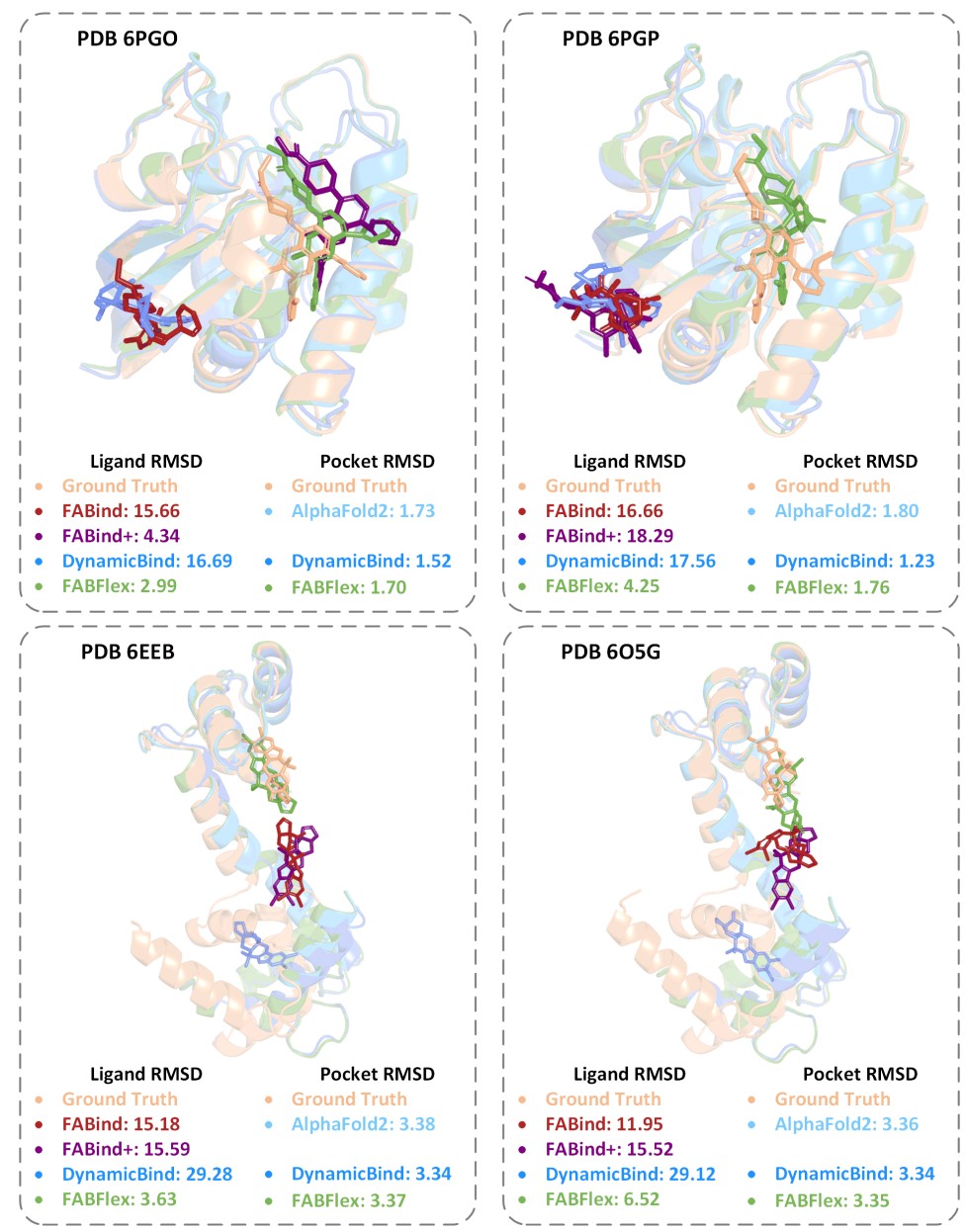

Figure 11: Four more case studies of PDB 6PGO, PDB 6PGP, PDB 6EEB, and PDB 6O5G.

cases, the predicted pockets are correctly located in the correct regions, centered around the binding sites of their respective holo ligands. This ensures that the predicted ligands are also correctly positioned within the pocket region.

## D  BROADER IMPACTS AND LIMITATIONS

**Broader impacts.** In the current landscape, with the vast number of known proteins and small molecules, a high-efficiency molecular docking method is essential for analyzing and processing large-scale molecule-protein interactions within a limited timeframe, aiding in the discovery of potential drug candidates with therapeutic value. Therefore, the exploration and development of fast and accurate docking methods in real-world scenarios are crucial to advancing AI-driven technologies in drug discovery.

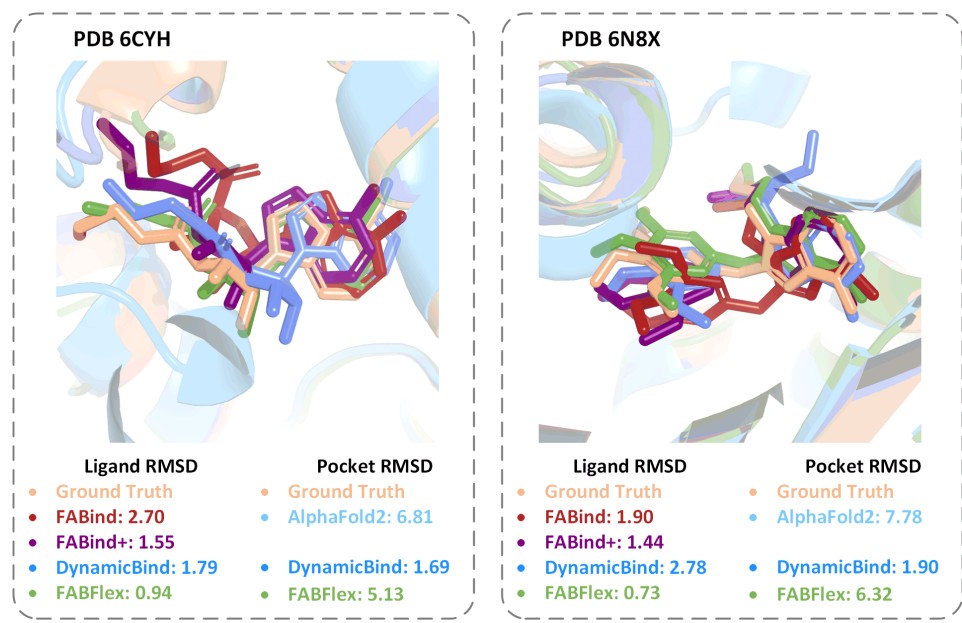

Figure 12: Two more case studies of PDB 6CYH and PDB 6N8X.

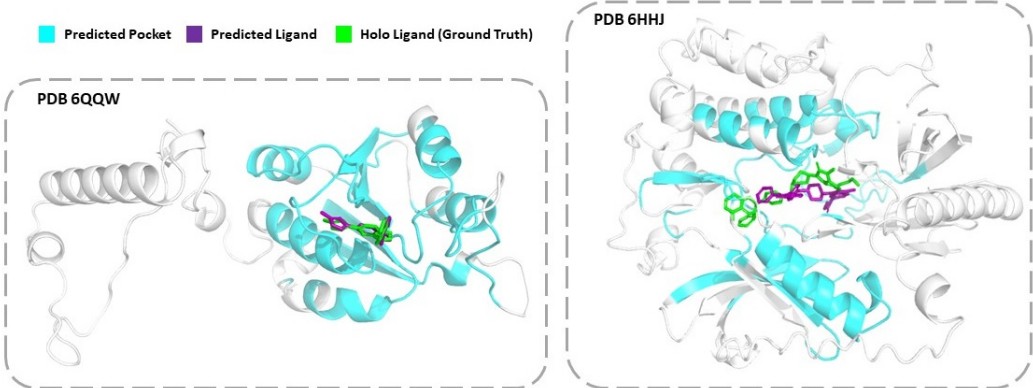

Figure 13: Two case studies of PDB 6QQW and PDB 6HHJ to showcase pocket prediction.

**Limitations.** There still are several limitations in this study. Firstly, the performance of pocket structures does not surpass that of the advanced method DynamicBind. This discrepancy motivates us to incorporate advantages of the sampling-based strategy to enhance the results. Additionally, deep learning-based methods tend to produce structures with more clashes compared to traditional docking software. To address this, a post-optimization method could be implemented to mitigate such issues (Stärk et al., 2022). Alternatively, integrating biological or chemical constraints into model design and optimization might also help alleviate these problems (Jiang et al., 2024).

