# OpenReview forum: "Fast and Accurate Blind Flexible Docking"
_ICLR.cc/2025/Conference — ICLR 2025 Poster_

### Official Review · Reviewer_R36f · 2024-10-22

**Soundness:** 4
**Presentation:** 4
**Contribution:** 3
**Rating:** 8
**Confidence:** 5

**Summary:**

This work proposes FABFlex, a regression-based model for fast and accurate docking of protein-ligand pairs. The pipeline of blind flexible docking is decomposed of three stages: 1) predicting the binding pockets, 2) predicting the holo structure of ligand and 3) predicting the holo structure of receptor pockets. An iterative update mechanism is utilized for continuous structure refinement. Solid experimental results verify the superiority of FABFlex in terms of ligand prediction, pocket prediction and inference efficiency.

**Strengths:**

1. At a time when diffusion is prevalent, the work verifies the efficacy of iterative update mechanism for structure refinement, which achieves extremely high efficiency and better experimental results.
2. Very solid experimental results are reported to show the superiority of FABFlex, including multiple tasks of interest in blind docking and various strong baselines.

**Weaknesses:**

1. Although good experimental results were achieved, the innovation of this work is relatively insufficient (e.g., using existing network architectures and training objectives), which is only reflected in the proposed iterative update mechanism.
2. In some tasks, the results of FABFlex are worse than those of some models based on protein rigidity prior (e.g., pocket prediction performance in Table 2). This anomaly requires further explanation.

**Questions:**

1. In Table 2 we see that FABind and FABind+ achieve best results on some metrics, which seems somewhat counterintuitive since they are tailored based on the assumption of protein rigidity. Could you please give futher explanations of this phenomenon?
2. In Table 3, the ablation of "iterative internally" shows better results on mean and median of ligand RMSD. I wonder if there is room for further improvement in protein-ligand interaction modeling.

**Details Of Ethics Concerns:**

No concerns.

---

> ### Author Response · Authors · 2024-11-21
> **Response to reviewer R36f (Part 1/2)**
>
> We are deeply grateful to the reviewer's valuable time and effort in reviewing our paper.
>
> >**W1:** Although good experimental results were achieved, the innovation of this work is relatively insufficient (e.g., using existing network architectures and training objectives), which is only reflected in the proposed iterative update mechanism.
>
> **A:** Thanks for your thoughtful comments. We believe that the innovation of a work should be considered from multiple perspectives. We summarize our contribution of this work as follows:
>
> - **Scenario perspective:** Our research targets **blind flexible docking, a task that is both essential and highly challenging** as it aligns more closely with real-world molecular docking. This scenario is still in its early state of development, and most existing studies focus on simplified docking scenario with maintaining rigid protein. Our work introduces a straightforward yet efficient framework consisting of a pocket prediction module, a ligand docking module, and a pocket docking module, achieving accurate docking performance with high computational efficiency compared to existing diffusion-based flexible docking models.
>
> - **Technical perspective:** The key contribution of our work is **the first work to explore the potentials of regression-based docking methods for faster and more efficient flexible docking**. To our knowledge, current flexible docking methods, such as DynamicBind, ReDock, PackDock, and NeuralPlexer, primarily rely on diffusion-based models with sampling strategy, which often leads to long runtime. However, regression-based methods like TankBind, E3Bind, and FABind demonstrate high computational speed but are constrained by the rigid protein assumption. As far as we know, **regression-based paradigm has not been explored or discussed for flexible docking scenario, and our work bridges this gap** by offering a fresh perspective to solve this problem.
>
> - **Design perspective:** We believe that the core motivation of this work is the design philosophy that empowers the regression-based framework to handle protein flexibility, and the FABind layer serves solely as base to construct our model. **Compared to the original FABind model, which is constrained by the rigid protein assumption and unable to handle protein flexibility, FABFlex currently has capacity to overcome this limitation and predict accurate docking structures.** We believe that this is attributed to our design philosophy: we clearly decompose the blind flexible docking into three subtasks; we design specific module to handle corresponding subtask; and we introduce an iterative update mechanism to simulate the interactions during the docking process. All of these points work collaboratively to achieve our goal: a regression-based model to tackle blind flexible docking effectively and efficiently.
>
> Overall, we believe our work provides new insights and serves as an inspiration for further exploration and discussion on whether the regression paradigm is suitable for flexible docking.
>
> >**W2:** In some tasks, the results of FABFlex are worse than those of some models based on protein rigidity prior (e.g., pocket prediction performance in Table 2). This anomaly requires further explanation.
> >**Q1:** In Table 2 we see that FABind and FABind+ achieve best results on some metrics, which seems somewhat counterintuitive since they are tailored based on the assumption of protein rigidity. Could you please give futher explanations of this phenomenon?
>
> **A:** Thanks for your valuable question. The performance in Table 2 is **mainly to indicate that FABFlex consistently outperforms P2Rank** in performance and efficiency regardless of whether the input is an apo protein or a holo protein, while maintaining the comparable performance with FABind and FABind+ as they share the same architecture of the pocket prediction module and binary classification strategy for identifying binding pocket sites. The differences between FABFlex and FABind, FABind+ lie in their training objectives, training strategies, and the subsequent docking modules. Specifically, FABind+ introduces an additional radius loss to supervise the dynamic pocket radius prediction compared FABind. To handle flexible docking, FABFlex introduces a pocket docking module and an iterative update mechanism, along with a pocket coordinate loss to supervise the predicted pocket conformation. Besides, FABFlex adopts a partial teacher-forcing training strategy rather than FABind+'s complete teacher-forcing. **Note that, these differences do not influence the core essence of using such pocket prediction module and binary classification to identify binding pocket residues.** To provide intuitive visualization about our pocket prediction, we supplement additional case studies in Figure 12 of the revised paper.

---

> ### Author Response · Authors · 2024-11-21
> **Response to reviewer R36f (Part 2/2)**
>
> >**Q2:** In Table 3, the ablation of "iterative internally" shows better results on mean and median of ligand RMSD. I wonder if there is room for further improvement in protein-ligand interaction modeling.
>
> **A:** Thanks for your careful observation. While the ablation of "iterative internally" demonstrates better results on mean and median of ligand RMSD, it shows a notable decline in the percentage of ligand RMSD < 2 Å, which is an important metric for evaluating the success rate of docking. To further investigate whether there is room to enhance this aspect, we have supplemented an additional experimental result for **"iterative both," which combines iterative exchanges between the ligand and pocket docking modules with internal iterations within each module (i.e., FABFlex + "iterative internally")**.  The results are shown as follows:
>
> | Method | Ligand RMSD (Å) |  |          | Pocket RMSD (Å) |          | Avg. Runtime (s) |
> |:-|:-:|:-:|:-:|:-:|:-:|:-:|
> |                      | Mean            | Median   | < 2Å (%) | Mean            | Median   |                  |
> | FABFlex              | 5.44            | 2.96     | 40.59    | 1.10            | 0.63     | 0.49             |
> | Iterative Internally | 5.35            | 2.94     | 35.31    | 1.42            | 0.88     | 2.05             |
> | Iterative Both       | 5.80            | 3.02     | 33.99    | 1.73            | 0.94     | 12.07            |
>
> It can be observed that "iterative both" leads to worse performance and longer runtime compared to both FABFlex and "iterative internally". This may be because we only compute gradients during the final iteration of the iterative update, and excessive iterations without gradient updates could accumulate errors or overcorrections, reducing overall performance and efficiency.
>
> The core motivation and contribution of this work are to provide the first exploration to discuss the potentials of employing regression-based paradigm for more efficient flexible docking. Thus, we think that current iterative update mechanism is one possible way to model the ligand-protein interaction, though it may not be the optimal way. We believe that the question of how to model ligand-protein interactions in flexible docking, even in more complicated scenarios like multi-pocket flexible docking, is a core challenge, remaining an open problem that awaits further exploration and discussion in future work.

---

> > ### Comment · Reviewer_R36f · 2024-11-22
> >
> > I have reviewed the authors' responses to all reviewers and greatly appreciate their deep understanding of the field and the extensive experiments conducted to address the concerns. The authors' replies have resolved my concerns, and I acknowledge the contribution of FABFlex to explore the potentials of regression-based docking methods for faster and more efficient flexible docking. I will raise my score.
> >
> > Best wishes!

---

> > > ### Author Response · Authors · 2024-11-22
> > > **Thank you**
> > >
> > > Dear Reviewer R36f,
> > >
> > > We sincerely appreciate your great support and insightful comments, which mean a lot to us! Moreover, following your valuable suggestions, we have made revisions to enhance the quality of our paper.
> > >
> > > Thank you once again for your valuable time and efforts!
> > >
> > > Best regards,
> > >
> > > Authors of # 7007

---

### Official Review · Reviewer_w792 · 2024-10-29

**Soundness:** 3
**Presentation:** 3
**Contribution:** 2
**Rating:** 6
**Confidence:** 4

**Summary:**

This paper introduces FABFlex, a framework for fast and accurate blind flexible docking. It consists a pocket identification module (blind), a ligand conformation prediction module (docking), and a protein flexibility modeling module (flexible).  FABFlex achieves sota results on  blind flexible docking benchmark, and is 208x faster than previous sampling-based deep learning methods.

**Strengths:**

1. The paper is well-written, with clearly designed figures and thorough explanations of each component.

2. The experiments, ablation studies, and visualizations are comprehensive and well-detailed.

3.  The framework demonstrates strong performance and significantly faster inference times compared to sampling-based approaches.

**Weaknesses:**

1. The primary concern is that this work appears to be a direct application of the FABind series. It just introduces an additional pocket conformation prediction module to handle the flexible docking setting, which limits the overall contribution and novelty of the paper.

**Questions:**

1. Why does this paper focus solely on blind (global) flexible docking, instead of exploring pocket-based (local) flexible docking? It seems feasible to adapt the framework to a local flexible docking setting and compare it against models such as DiffDock-Pocket[1], ReDock[2], FlexPose[3], and DiffBindFR[4].

[1] Plainer, Michael, et al. "DiffDock-Pocket: Diffusion for Pocket-Level Docking with Sidechain Flexibility." NeurIPS 2023 Workshop on New Frontiers of AI for Drug Discovery and Development.

[2] Huang, Yufei, et al. "Re-Dock: Towards Flexible and Realistic Molecular Docking with Diffusion Bridge." Forty-first International Conference on Machine Learning.

[3] Dong, Tiejun, et al. "Equivariant flexible modeling of the protein–ligand binding pose with geometric deep learning." Journal of Chemical Theory and Computation 19.22 (2023): 8446-8459.

[4] Zhu, Jintao, et al. "DiffBindFR: an SE (3) equivariant network for flexible protein–ligand docking." Chemical Science 15.21 (2024): 7926-7942.

---

> ### Author Response · Authors · 2024-11-21
> **Response to reviewer w792 (Part 1/2)**
>
> We are deeply grateful to the reviewer for dedicating valuable time and effort to review our paper.
> >**W1:** The primary concern is that this work appears to be a direct application of the FABind series. It just introduces an additional pocket conformation prediction module to handle the flexible docking setting, which limits the overall contribution and novelty of the paper.
>
> **A:** Thanks for your insightful comments. We summarize our contribution as follows:
> - From scenario perspective, **we focus on the blind flexible docking**, a crucial yet challenging scenario that better reflects the realistic molecular docking process. While most existing studies primarily focus on simplified docking scenario with the rigid protein assumption, only a few have explored this more complex problem. Our work proposes a simple yet effective framework based on fundamental FABind layers, including a pocket prediction module, a ligand docking module, and a pocket docking module, to tackle this problem. Our model achieves strong performance with maintaining high computational efficiency, surpassing the computationally intensive methods currently available.
> - From technique perspective, the core contribution of our work is **the first work to explore the potentials of regression-based docking methods for accurate and more efficient flexible docking**. As far as we know, existing flexible docking methods such as DynamicBind, ReDock, PackDock and NeuralPlexer, predominantly rely on diffusion models with sampling strategy to handle protein flexibility, which often results in low computational efficiency. In contrast, regression-based methods such as TankBind, E3Bind and FABind excel in computational speed yet are limited by the rigid protein assumption. **To date, there has been little effort to investigate the feasibility of regression-based paradigms for flexible docking scenarios**. Our work seeks to bridge this gap, offering a new technical route to discuss the problem of flexible docking.
> - From design perspective, FABind layer is just a specific E(3)-equivariant graph neural network (GNN) designed for ligand-protein heterogenous graph. FABind, FABind+ and FABFlex all utilize this specific GNN to construct their models. Yet, compared to FABind and FABind+ limited by rigid protein assumption, FABFlex has capacity to handle protein flexibility. **We make necessary modification with clear design philosophy to make it possible to achieve our goal: a regression-based methods for blinding flexible docking**. Specifically, we clearly decompose the blind flexible docking into three concrete subtasks, each handled by a dedicated module. We design an iterative update mechanism to model the docking interactions effectively, and its effect can be observed from the case studies in Figure 4 and Figure 9. Generally, we think that the FABind layer serves solely as the base, and these thoughtful yet straightforward enhancements are the key to empower our model to perform fast and accurate blind flexible docking in a regression-based mannar.
>
> We review and summarize a table to compare the existing studies from various angles for further discussion as follows:
> |Method|Rigid or Flexible|Sampling required?| External pocket tool required? | Regression or Diffusion | Avg. Runtime < 1s |
> |:--|:-:|:-:|:-:|:-:|:-:|
> |TankBind[1]|Rigid|No|Yes|Regression|Yes|
> |E3Bind[2]|Rigid|No|Yes|Regression|Yes |
> |FABind [3]|Rigid|No|No|Regression|Yes|
> |FABind+ [4]|Rigid|No|No|Regression| Yes |
> |DiffDock [5]|Rigid|Yes|No|Diffusion|No|
> |DynamicBind [6]| Flexible|Yes|No|Diffusion | No |
> |NeuralPLexer [7]| Flexible | Yes | No | Diffusion | No |
> | FABFlex | Flexible | No | No | Regression | Yes |
>
> It can be observed that **FABFlex stands out among existing methods by leveraging a regression-based approach to tackle flexible docking without relying on sampling strategy or external pocket detection tools, achieving a fast runtime**. Moreover, to intuitively clarify the efficiency drawbacks of existing diffusion-based flexible docking models, we provide additional experimental results about the number of samplings to further analyze:
>
> |Methods|Ligand RMSD (Å)|||Pocket RMSD (Å)||Avg. Runtime (s)|
> |:-|:-:|:-:|:-:|:-:|:-:|:-:|
> ||Mean↓|Median↓|< 2Å (%)↑|Mean↓|Median↓||
> |DynamicBind(1)|6.26|3.45|27.15|0.84|0.59|22.04|
> |DynamicBind(10)|6.21|3.41|27.48|0.84|0.57|47.22|
> |DynamicBind(40)|6.19|3.16|33.00|0.84|0.58|102.12|
> |FABFlex|5.44|2.96|40.59|1.10|0.63|0.49|
>
> We can observe that there is a performance degradation when using fewer number of samplings. Even DynamicBind(1) only sample once, its runtime is still much slower than that of regression-based models. These observations reflect the inherent trade-off between performance and efficiency in diffusion-based methods, where the runtime grows with the number of samplings. In summary, we believe that our work provides an exploration to discuss the feasibility of regression-based methods for efficient flexible docking.

---

> ### Author Response · Authors · 2024-11-21
> **Response to reviewer w792 (Part 2/2)**
>
> >**Q1:** Why does this paper focus solely on blind (global) flexible docking, instead of exploring pocket-based (local) flexible docking? It seems feasible to adapt the framework to a local flexible docking setting and compare it against models such as DiffDock-Pocket, ReDock, FlexPose, and DiffBindFR.
>
> **A:** Thanks for your valuable question. Both pocket-based (local) flexible docking and blind (global) flexible docking are important research directions, each with its own strengths and applicable scenarios. Pocket-based flexible docking is well-suited for cases where reliable prior knowledge of the binding site or pocket sidechain is available. Methods such as DiffDock-Pocket, ReDock, FlexPose, and DiffBindFR have demonstrated the value of this scenario in flexible docking research. Notably, most of these methods are based on diffusion models. Adapting the regression-based paradigm to this scenario presents a promising direction for future work.
>
> In this work, we focus on blind flexible docking, because it represents a realistic and much more challenging problem, where the binding pocket sites are unknown and protein retains its flexible nature during the docking process. This direction is still in its preliminary development, but the importance has been proved, works such as DynamicBind [6] published in Nature Communication and NeuralPLexer [7] published in Nature Machine Intelligence. As a promising and challenging direction, we aim to tackle this problem with our little contribution. This setting is applicable to the scenario to evaluate molecule-protein interactions without prior knowledge of the binding sites and protein conformations. Thanks to AlphaFold series, we can now easily obtain the accurate 3D structural predictions of proteins from their amino acid sequences as apo protein conformations. And the ETKDG algorithm in RDKit can generate apo ligand structures that satisfy both biological and chemical constraints from molecules' SMILES sequences. Thus, these advancements have naturally guided us to focus on the blind flexible docking to address the limitation of rigid protein assumption in many existing studies.
>
> [1] Lu W, Wu Q, Zhang J, et al. Tankbind: Trigonometry-aware neural networks for drug-protein binding structure prediction[J]. Advances in neural information processing systems, 2022, 35: 7236-7249.
> [2] Zhang Y, Cai H, Shi C, et al. E3Bind: An End-to-End Equivariant Network for Protein-Ligand Docking[C]//The Eleventh International Conference on Learning Representations, 2023.
> [3] Pei Q, Gao K, Wu L, et al. FABind: fast and accurate protein-ligand binding[C]//Proceedings of the 37th International Conference on Neural Information Processing Systems. 2023: 55963-55980.
> [4] Gao K, Pei Q, Zhu J, et al. FABind+: Enhancing Molecular Docking through Improved Pocket Prediction and Pose Generation[J]. arXiv preprint arXiv:2403.20261, 2024.
> [5] Corso G, Stärk H, Jing B, et al. DiffDock: Diffusion Steps, Twists, and Turns for Molecular Docking[C]//The Eleventh International Conference on Learning Representations, 2023.
> [6] Lu W, Zhang J, Huang W, et al. DynamicBind: Predicting ligand-specific protein-ligand complex structure with a deep equivariant generative model[J]. Nature Communications, 2024, 15(1): 1071.
> [7] Qiao Z, Nie W, Vahdat A, et al. State-specific protein–ligand complex structure prediction with a multiscale deep generative model[J]. Nature Machine Intelligence, 2024, 6(2): 195-208.

---

> ### Author Response · Authors · 2024-11-22
> **Looking forward to your reply**
>
> Dear Reviewer w792,
>
> Thanks for your valuable time and great efforts in reviewing our work and providing insightful comments. We hope that our answers and general responses have helped to clarify the points discussed. Please let us know if there is anything else you require further information on or if there are any additional concerns you might have.
>
> Best regards,
>
> Authors of # 7007

---

> ### Comment · Reviewer_w792 · 2024-11-23
>
> Thank you for your response. I appreciate the authors’ detailed explanation and agree that the blind flexible docking scenario is indeed more challenging. However, including results for the pocket-based flexible docking setting would significantly enhance the paper for several reasons:
>
> 1.  FABFlex employs a two-stage process for blind flexible docking, with a pocket prediction stage followed by a docking stage using the predicted pocket. If the first stage is bypassed and the ground truth pocket is directly used, this effectively transforms the approach into the pocket-based flexible docking setting. Such an experiment would not represent a different experimental scenario but rather serve as an important ablation study to isolate the performance of the docking stage. Similar analyses, such as those in Table 2 (evaluating pocket prediction alone) and Table 3 (comparing the use of P2Rank for pocket prediction), already demonstrate the effectiveness of the pocket prediction module. Conducting a study where the pocket prediction stage is replaced with the ground truth pocket would naturally complement these analyses.
>
> 2.  Among the deep learning baselines discussed, only DynamiBind shares the same experimental setting as FABFlex. In contrast, the pocket-based flexible docking setting offers a broader range of baseline methods for comparison. Including this experiment would provide stronger evidence for the effectiveness of the regression-based docking approach in predicting accurate conformations and poses.
>
> 1. Demonstrating FABFlex's ability to handle both blind flexible docking and pocket-based flexible docking scenarios would significantly strengthen its impact and contribution. As a two-stage framework, FABFlex should inherently have the flexibility to adapt to different settings. Highlighting this capability would be a major advantage of the approach and elevate its overall significance.
>
>
> I greatly appreciate the authors' effort in writing this well-crafted paper and would be willing to increase my score if the suggested results are provided. I understand that obtaining these results may pose challenges, especially if it requires removing the partial teacher-forcing strategy and retraining the model, which could be a time-intensive process for a docking model.

---

> ### Author Response · Authors · 2024-11-25
> **Response to reviewer w792**
>
> >**Q:** including results for the pocket-based flexible docking setting would significantly enhance the paper
>
> **A:** Thanks to your insightful comments regarding pocket-based docking. We fully agree with you that pocket-based flexible docking is a crucial scenario, and  thanks for the suggestion that incorporating the experimental results of pocket-based flexible docking is beneficial to enhance the quality of our paper. Therefore, we conduct additional experiments where FABFlex is retrained for pocket-based docking scenario with prior pocket knowledge. **We follow your suggestion: the pocket prediction module is bypassed, and the partial teacher-forcing strategy is removed to adapt FABFlex model to pocket-based docking, where we feed the ground truth pocket amino acids and try to predict the pocket structure and ligand structure**. Moreover, we supplement the evaluations of FABind and FABind+ in pocket-based docking scenario, and we include the results reported in DiffDock-Pocket paper [1] and ReDock paper [2] for comparison. Due to time limitation, the results are referenced from their respective papers, we will reproduce their methods in the future for a fairer comparison. But we believe their own reported results would be more convincing. These results are presented as follows:
>
> | Methods               | Ligand RMSD (Å) |          |          |          | Pocket RMSD (Å) |          | Avg. Runtime |
> |:-------------|:------------:|:----------:|:----------:|:----------:|:-----------:|:----------:|:------------:|
> |                       | Mean ↓         | Median ↓ | < 2Å (%) ↑ | < 5Å (%) ↑ | Mean ↓         | Median ↓ | (s)          |
> | FABind                | 4.47            | 3.23     | 23.76    | 67.66    | -               | -        | 0.10         |
> | FABind+               | 4.21            | 2.66     | 36.63    | 69.97    | -               | -        | 0.14         |
> | DiffDock-Pocket(10)   | -               | 2.60     | 41.00    | -        | -               | -        | 17           |
> | DiffDock-Pocket(40)   | -               | 2.60     | 41.70    | -        | -               | -        | 61           |
> | ReDock(10)            | -               | 2.50     | 39.00    | 74.80    | -               | -        | 15           |
> | ReDock(40)            | -               | 2.40     | 42.90    | 76.40    | -               | -        | 58           |
> | FABFlex               | 3.45            | 2.57     | 42.24    | 75.25    | 0.93            | 0.66     | 0.47         |
>
> It can be observed that FABFlex is also effective in pocket-based docking scenario, achieving a **Ligand RMSD < 2Å of 42.24%, outperforming FABind of 23.76%, FABind+ of 36.63%, DiffDock-Pocket(10) of 41.0%, DiffDock-Pocket(40) of 41.70%, ReDock(10) of 39.0%, and comparable to ReDock(40) of 42.90%**. Furthermore, FABFlex achieves a Pocket RMSD Mean of 0.93Å, demonstrating its ability to model protein flexibility. On Ligand RMSD < 5A, FABlex achieves 75.25%, comparable to ReDock(40), surpassing FABind, FABind+ and ReDock(10). Notably, **the average runtime of FABFlex is only 0.47 seconds, which is considerably faster than DiffDock-Pocket(40) and ReDock(40) (more than 100x times faster)**, indicating the strong efficiency of our FABFlex. These results suggest that the regression-based paradigm has the potential and capacity to handle protein flexibility in both blind and pocket-based docking scenarios. These results have been supplemented in the Appendix C.9 of our revised paper.
>
> [1] Plainer M, Toth M, Dobers S, et al. Diffdock-pocket: Diffusion for pocket-level docking with sidechain flexibility[J]. 2023.
> [2] Huang Y, Zhang O, Wu L, et al. Re-Dock: Towards Flexible and Realistic Molecular Docking with Diffusion Bridge[C]//Forty-first International Conference on Machine Learning.

---

> > ### Comment · Reviewer_w792 · 2024-11-25
> >
> > Thank you for your response. The results demonstrate that the docking module of FABFlex is effective. I have raised the score.

---

> > > ### Author Response · Authors · 2024-11-25
> > > **Thank you**
> > >
> > > Dear Reviewer w792,
> > >
> > > We sincerely appreciate your great support and insightful comments, which mean a lot to us! Moreover, following your valuable suggestions, we have made revisions to enhance the quality of our paper.
> > >
> > > Thank you once again for your valuable time and efforts!
> > >
> > > Best regards,
> > >
> > > Authors of # 7007

---

### Official Review · Reviewer_H5gN · 2024-11-03

**Soundness:** 3
**Presentation:** 3
**Contribution:** 2
**Rating:** 6
**Confidence:** 5

**Summary:**

In this paper, the authors introduce FABFlex, a regression-based flexible blind docking model. FABFlex uses the EGNN architecture, with FABind as its main layer. The model is composed of three modules: (1) the pocket prediction module, (2) the ligand docking module, and (3) the pocket docking module. FABFlex focuses on finding holo structures to enable flexible docking. Compared to other models, FABFlex demonstrates accuracy with a higher percentage of ligands achieving RMSD < 2 Å and RMSD < 5 Å, and it is also accurate in identifying holo pockets based on RMSD metrics.

**Strengths:**

- The paper presents all details clearly and comprehensibly.
- This is the one of first applications of regression-based flexible blind docking.
- It employs an interesting multi-task approach, addressing more than a single task simultaneously.
- The use of flexibility, rather than just rigidity, in docking, combined with a regression approach, makes the paper compelling.
- The pipeline provides a systematic, data-driven approach to molecular docking.
- The experiments are detailed and clearly presented, with comprehensive benchmarking against other models.
- The code is openly shared (though the README is empty; see weaknesses).

**Originality:** FABFlex is innovative as a regression-based model for flexible docking. It also stands out for its multi-task capability, predicting not only apo structures but also holo-structures.

**Quality:** The paper's quality is highlighted by its comprehensive benchmarking against other models. Additionally, it goes beyond ligand RMSD to include pocket RMSD analyses, further enhancing the study’s depth.

**Clarity:** The paper is well-written grammatically, with equations that are scientifically clear, readable, and easy to follow.

**Weaknesses:**

- This study closely resembles the FABind[1,2] approach in both training and inference, with many of the techniques used already present in FABind. Consequently, flexible docking appears somewhat overshadowed by FABind.
- It is not specified whether protein preprocessing is used for inference runtime, affecting the runtime comparison.
- Using binary classification to identify pocket regions could limit flexibility at the atomic level; a more adaptable approach may be beneficial.
- The study relies solely on the RMSD metric, which does not always ensure plausible structures. Metrics such as semi-empirical binding affinity, protein-ligand steric clashes, and ligand strain energy, as suggested by [3,4,5,6,7], should be considered.
- Although proteins and ligands are visually classified as apo or holo, this classification does not guarantee bioactive, chemical, or physical plausibility. Including metrics like binding affinity, steric clashes, and ligand strain energy would provide greater insight, as demonstrated in studies like Posebuster[3], PoseCheck[4], PoseBench[5], CompassDock[6], and PLINDER[7].
- The binding affinity, protein-ligand steric clashes, and ligand strain energy for the processed PDBBind’s apo and holo structures were not examined.
- While RMSD may increase due to holo pocket region flexibility, the focus should instead be on binding affinity, steric clashes, and ligand strain energy, but the study only emphasizes RMSD.
- It is unclear if other DL-based models in the benchmark used the same timesplit or were retrained for comparability.
- If RMSD is the chosen metric, timesplit information leakage may occur, as described in the PLINDER study. Clarifying timesplitting methods would strengthen the study.
- As a FABind user, I found the reported runtime of 0.12 sec inaccurate. When protein preprocessing is included, runtime is closer to 10-15 seconds, with 0.12 sec applying only to ligand conformation prediction. Based on FABind, I expect FABFlex’s runtime to exceed the stated 0.49 sec with protein preprocessing included.
- Although FABFlex claims faster runtime than DiffDock, normalizing per-ligand runtime suggests DiffDock may be more efficient, as DiffDock samples ~40 ligand conformations in ~2 sec per conformation, while FABFlex takes ~10-15 sec for one prediction.

**Quality:** The RMSD metric may lack biological, chemical, or physical relevance. For benchmarking, it would be better to include up-to-date metrics like semi-empirical binding affinity, protein-ligand steric clashes, and ligand strain energy.

**Clarity:** The criteria for selecting the values of alphas in the loss function are not clearly explained.

**Reproducibility:** Although the code is open source, the README section is empty, and usage instructions are not provided. If the README were clarified, I could test and reassess the code’s reproducibility. Additionally, the conda environment in the YAML file is named "FABind," raising anonymity concerns regarding double-blind review, as it suggests potential overlap with FABind’s authors.

**Minor Mistake**

"Subsequent" is used twice:
> "The predicted pocket sites by pocket prediction module enable the subsequent subsequent ligand"

### **References**
[1] Qizhi Pei, Kaiyuan Gao, Lijun Wu, Jinhua Zhu, Yingce Xia, Shufang Xie, Tao Qin, Kun He, Tie-Yan Liu, Rui Yan. FABind: Fast and Accurate Protein-Ligand Binding. arXiv preprint arXiv:2310.06763, 2023.

[2] Kaiyuan Gao, Qizhi Pei, Jinhua Zhu, Kun He, Lijun Wu. FABind+: Enhancing Molecular Docking through Improved Pocket Prediction and Pose Generation. arXiv preprint arXiv:2403.20261, 2024.

[3] Martin Buttenschoen, Garrett M Morris, and Charlotte M Deane.  Posebusters:  Ai-based docking methods fail to generate physically valid poses or generalise to novel sequences. Chemical Science, 15(9):3130–3139, 2024.

[4] Charles Harris, Kieran Didi, Arian R Jamasb, Chaitanya K Joshi, Simon V Mathis, Pietro Lio, and Tom Blundell. Benchmarking generated poses: How rational is structure-based drug design with generative models? arXiv preprint arXiv:2308.07413, 2023.

[5] Alex Morehead, Nabin Giri, Jian Liu, Jianlin Cheng. Deep Learning for Protein-Ligand Docking: Are We There Yet? arXiv preprint arXiv:2405.14108, 2024.

[6] Ahmet Sarigun, Vedran Franke, Bora Uyar, Altuna Akalin. CompassDock: Comprehensive Accurate Assessment Approach for Deep Learning-Based Molecular Docking in Inference and Fine-Tuning. arXiv:2406.06841, 2024.

[7] Janani Durairaj et al. PLINDER: The protein-ligand interactions dataset and evaluation resource. bioRxiv, 2024

**Questions:**

- Do you account for protein preprocessing during inference? Based on my experience with FABind, preprocessing typically takes around 10-15 seconds.
- Does the flexibility in your model prevent steric clashes between protein and ligand?
- Did you use the same timesplitting approach for training other DL-based methods as you did for FABFlex?
- Did you experiment with different pocket site radii, such as 10 Å or 5 Å, in addition to 20 Å?
- Are the alpha values learnable, or do you use a predetermined approach for them?

---

> ### Author Response · Authors · 2024-11-21
> **Response to reviewer H5gN (Part 1/5)**
>
> We sincerely appreciate the reviewers' valuable time and effort in reviewing our paper.
> >**W1:** This study closely resembles the FABind[1,2] approach in both training and inference, with many of the techniques used already present in FABind. Consequently, flexible docking appears somewhat overshadowed by FABind.
>
> **A:** Thanks for your insightful comments. While it is true that FABFlex is built upon the fundamental FABind layer, the core contribution and motivation of this work are **the first to explore the potentials of the regression-based framework for flexible docking**. Existing flexible docking methods predominately rely on diffusion models with sampling strategy, while existing regression-based docking methods are typically constrained by the assumption of protein rigidity. This leaves a critical gap in exploring applicability of regression-based framework for efficient flexible docking, which our work aims to address. From experimental results of our study, we demonstrate that through thoughtful task decomposition, well-designed modules, and meticulous training, a regression-based framework can be effectively applied to tackle the challenging blind flexible docking task.
>
> Besides, we acknowledge that FABind is a well-executed and valuable work, and our study builds upon its foundation. However, rather than suggesting that flexible docking is overshadowed by FABind, we view our work as an exploration of the regression-based framework, showcasing its utility and demonstrating its versatility in addressing the complex challenges of flexible docking. Just as the Transformer architecture has inspired numerous adaptations to address various scenarios without diminishing its significance, FABind serves as a strong foundation that enables further innovation, such as ours, to explore and extend its potential and applicability in new flexible docking.

---

> ### Author Response · Authors · 2024-11-21
> **Response to reviewer H5gN (Part 2/5)**
>
> >**W2:** It is not specified whether protein preprocessing is used for inference runtime, affecting the runtime comparison.
> >**W10:** As a FABind user, I found the reported runtime of 0.12 sec inaccurate. When protein preprocessing is included, runtime is closer to 10-15 seconds, with 0.12 sec applying only to ligand conformation prediction. Based on FABind, I expect FABFlex’s runtime to exceed the stated 0.49 sec with protein preprocessing included.
> >**W11:** Although FABFlex claims faster runtime than DiffDock, normalizing per-ligand runtime suggests DiffDock may be more efficient, as DiffDock samples ~40 ligand conformations in ~2 sec per conformation, while FABFlex takes ~10-15 sec for one prediction.
> >**Q1:** Do you account for protein preprocessing during inference? Based on my experience with FABind, preprocessing typically takes around 10-15 seconds.
>
> **A:** Thanks for your valuable comments. As several questions and concerns are raised regarding the inference runtime, we explain them collectively here. Firstly, we clarify that the reported average runtime is the specific docking inference time, which begins with the  input apo ligand and apo protein conformations, and ends with the predicted holo ligand and holo protein structures. This runtime **excludes any preprocessing steps or post-optimizing conformations**. Moreover, this is also **a common practice for comparing docking computational efficiency used in many existing studies**, such as TankBind [1], E3Bind [2], DiffDock [3], and FABind [4], in which they also compare the docking runtime not including preprocessing time. Consequently, we think it is a fair comparison about the docking computational efficiency.
>
> Then, we explain the arguments about the preprocessing time. Preprocessing primarily involves steps such as initializing the ligand structure from its SMILES sequence, generating the protein conformation from its amino acid sequence (e.g., using AlphaFold2), and constructing features for input graphs. Since these steps are external to the docking process itself, they are not included in the runtime comparison. In our study, apo protein conformations are generated using AlphaFold2 and provided as inputs to all methods to ensure consistency. If the preprocessing time of AlphaFold2 are included, **it would uniformly affect all methods, as they would also require the same initial protein conformations**. Consequently, including this time would not impact the relative comparison of docking runtime, because all methods add a same time, but it would instead render the comparison redundant.
>
>
> Next, we clarify the runtime of FABind and FABFlex. As discussed above, the reported runtimes of 0.12 seconds for FABind and 0.49 seconds for FABFlex reflect the docking computational time only and do not include preprocessing. The reviewer notes that FABind takes approximately 12 seconds; this discrepancy likely arises because the reviewer measured the runtime including the execution of `inference_preprocess_mol_confs.py`, `inference_preprocess_protein.py`, and `fabind_inference.py`. These scripts encompass the entire preprocessing pipeline, including data and feature preparation as well as docking, which explains the longer observed runtime. **If the reviewer uses the `test_fabind.py` script to measure the runtime of all test cases, it would align with the reported 0.12 seconds. FABFlex follows a similar runtime calculation as existing work.**
>
> Lastly, we have supplemented additional experimental results of the number of samplings in Appendix C.2 to further discuss the efficiency of diffusion-based models, including DynamicBind(1), DynamicBind(10), and DynamicBind(40):
>
> | Methods | Ligand RMSD (Å) | | | Pocket RMSD (Å) | | Avg. Runtime (s) |
> |:--|:--:|:--:|:--:|:--:|:--:|:--:|
> | | Mean ↓ | Median ↓ | < 2Å (%) ↑ | Mean ↓ | Median ↓ | |
> | DynamicBind(1) | 6.26 | 3.45 | 27.15 | 0.84 | 0.59 | 22.04 |
> | DynamicBind(10) | 6.21 | 3.41 | 27.48 | 0.84 | 0.57 | 47.22 |
> | DynamicBind(40) | 6.19 | 3.16 | 33.00 | 0.84 | 0.58 | 102.12 |
> | FABFlex | 5.44 | 2.96 | 40.59 | 1.10 | 0.63 | 0.49 |
>
> It can be observed that: Firstly, reducing the number of samplings leads to a degradation in the docking performance of diffusion-based models; Secondly, even with a single sampling iteration, as seen in DynamicBind(1), the docking time exceeds that of regression-based methods such as TankBind, FABind, and FABFlex. These observations highlight that the primary limitation of diffusion-based methods lies in their computational efficiency.
>
> [1] Lu W, Wu Q, Zhang J, et al. Tankbind: Trigonometry-aware neural networks for drug-protein binding structure prediction.
> [2] Zhang Y, Cai H, Shi C, et al. E3Bind: An End-to-End Equivariant Network for Protein-Ligand Docking.
> [3] Corso G, Stärk H, Jing B, et al. DiffDock: Diffusion Steps, Twists, and Turns for Molecular Docking.
> [4] Pei Q, Gao K, Wu L, et al. FABind: fast and accurate protein-ligand binding.

---

> ### Author Response · Authors · 2024-11-21
> **Response to reviewer H5gN (Part 3/5)**
>
> >**W3:** Using binary classification to identify pocket regions could limit flexibility at the atomic level; a more adaptable approach may be beneficial.
>
> **A:** Thanks for your insightful comments. Most of the docking methods, including TankBind, E3Bind, FABind, and DynamicBind, all operate on proteins at the residue level when identifying pocket regions, and our FABFlex model follows the same standard. Identifying pocket regions at the residue level is sufficient for defining the binding site, as the primary goal of pocket identification is to locate the **approximate binding region** to navigate the docking process. As shown in Table 2 of our paper (provided below), the performance of the pocket prediction module in FABFlex surpasses that of the widely used external pocket detection tool P2Rank, further demonstrating FABFlex’s effectiveness in locating binding pocket sites. Additionally, we have supplemented case studies in Appendix Figure 12 to intuitively showcase the pocket prediction by our FABFlex.
>
> | Methods | MAE (Å) $\downarrow$ | RMSE (Å) $\downarrow$ | EucDist (Å) $\downarrow$ |
> |:--|:--:|:--:|:--:|
> | P2Rank | 4.04 | 5.69 | 7.85 |
> | FABFlex | 3.29 | 4.83 | 6.59 |
>
> >**W8:** It is unclear if other DL-based models in the benchmark used the same timesplit or were retrained for comparability.
> >**W9:** If RMSD is the chosen metric, timesplit information leakage may occur, as described in the PLINDER study. Clarifying timesplitting methods would strengthen the study.
> >**Q3:** Did you use the same timesplitting approach for training other DL-based methods as you did for FABFlex?
>
> **A:** Thanks for your thoughtful comments. Yes, the dataset split strategy is crucial for ensuring fairness in comparison and avoiding potential information leakage. We use the PDBBind v2020 dataset for our experiments, and **adopt the same dataset splitting strategy as existing studies** such as TankBind, DiffDock, FABind, and DynamicBind. Specifically, **we use structures before 2019 for training and validation, and those after 2019 for testing**. Subsequently, Our preprocessing steps align with the pipeline used in DynamicBind. To ensure a fair comparison, all baseline methods are performed on this same preprocessed dataset. There is no data leakage from training set to test set.

---

> ### Author Response · Authors · 2024-11-21
> **Response to reviewer H5gN (Part 4/5)**
>
> > **W4:** The study relies solely on the RMSD metric, which does not always ensure plausible structures. Metrics such as semi-empirical binding affinity, protein-ligand steric clashes, and ligand strain energy, as suggested by [3,4,5,6,7], should be considered.
> > **W5:** Although proteins and ligands are visually classified as apo or holo, this classification does not guarantee bioactive, chemical, or physical plausibility. Including metrics like binding affinity, steric clashes, and ligand strain energy would provide greater insight, as demonstrated in studies like Posebuster[3], PoseCheck[4], PoseBench[5], CompassDock[6], and PLINDER[7].
> > **W6:** The binding affinity, protein-ligand steric clashes, and ligand strain energy for the processed PDBBind’s apo and holo structures were not examined.
> > **W7:** While RMSD may increase due to holo pocket region flexibility, the focus should instead be on binding affinity, steric clashes, and ligand strain energy, but the study only emphasizes RMSD.
> > **W12:** Quality: The RMSD metric may lack biological, chemical, or physical relevance. For benchmarking, it would be better to include up-to-date metrics like semi-empirical binding affinity, protein-ligand steric clashes, and ligand strain energy.
> > **Q2:** Does the flexibility in your model prevent steric clashes between protein and ligand?
>
> **A:** We appreciate your thoughtful comments and suggestions regarding the evaluation metrics. We explain these questions about the evaluation metrics collectively here. Molecular docking task aims at predicting the 3D conformations of protein-ligand complexes, and **RMSD is a clear and intuitive metric widely used in existing docking studies**, such as EquiBind [5], TankBind, E3Bind, DiffDock, FABind, and DynamicBind, to evaluate the deviation between predicted structures and ground truth structures, so that we choose RMSD as the core evaluation metric.
>
> Yet, we fully agree that incorporating additional metrics would provide a more comprehensive evaluation of docking quality. Therefore, we supplement additional experimental results of the binding affinity and clash score. Specifically, **we employ the MBP [6] that is a structure-based binding affinity prediction model, to evaluate IC50 and K value of the predicted structures to reflect their binding affinity**. IC50 represents the concentration of a ligand required to inhibit 50% of the protein's activity, while K Value reflects the dissociation constant, directly measuring the strength of the ligand-protein binding affinity. For both metrics, smaller values indicate stronger binding affinity and better docking quality. The experimental results are in Appendix C.8 of revised papar and shown as follows:
>
> | Methods | FABind | FABind+ | DynamicBind | FABFlex |
> |:--|:--:|:--:|:--:|:--:|
> | **IC50 (μM) ↓**  | 6.32 | 6.24 | 6.27 | 5.92 |
> | **K Value (μM) ↓** | 6.23 | 6.19 | 6.18 | 6.10 |
>
> It can be observed that FABFlex achieves the lowest IC50 (5.92 μM) and K Value (6.10 μM) among all methods, suggesting its potential to better predict biologically relevant interactions. Additionally, following DynamicBind, **we use the clash score as a metric to assess steric clashes between the ligand and protein**. It is calculated as the root-mean-square of van der Waals overlaps for all atom pairs within a distance of less than 4 Å. The clash score is defined as follows:
>
> $\text{clash score} = \sqrt{\frac{\sum_{i=0}^N \text{VdW overlop}^2}{N}},$
>
> where $N$ is the number of distances considered. The experimental results are supplemented in Appendix C.7 of revised paper and shown as follows:
>
> |Methods|Vina|Glide|Gnina|TankBind|FABind|FABind+|DiffDock|DynamicBind|FABFlex|
> |:-:|:-:|:-:|:-:|:-:|:-:|:-:|:-:|:-:|:-:|
> |**clash score ↓**|0.02|0.08|0.05|0.41|0.51|0.45|0.33|0.27|0.37|
>
> It can be observed that traditional docking software produces significantly lower clash scores, while deep learning-based docking methods tend to have higher clash scores. This underscores a key challenge of deep learning approaches: despite their enhanced docking accuracy, they frequently encounter issues with physical clashes. To our knowledge, addressing steric clashes in deep learning-based docking methods remains an open research question.
>
> [5] Stärk H, Ganea O, Pattanaik L, et al. Equibind: Geometric deep learning for drug binding structure prediction[C]//International conference on machine learning. PMLR, 2022: 20503-20521.
> [6] Yan J, Ye Z, Yang Z, et al. Multi-task bioassay pre-training for protein-ligand binding affinity prediction[J]. Briefings in Bioinformatics, 2024, 25(1): bbad451.

---

> ### Author Response · Authors · 2024-11-21
> **Response to reviewer H5gN (Part 5/5)**
>
> >**W13:** Clarity: The criteria for selecting the values of alphas in the loss function are not clearly explained.
> >**Q5:** Are the alpha values learnable, or do you use a predetermined approach for them?
>
> **A:** Thanks for your valuable comments. The Implementation configuration is presented in Table 5 of our paper, including the $\alpha$ values, which are hyperparameters predefined before training process. We make explanation of selection of $\alpha$ values as follows: Firstly, for $α_1^{cls}$, $α_1^{center}$, $α_2$, $α_4$, we refer to the setting used in FABind as {1.0, 0.05, 1.5, 1.0}, respectively. For $α_3$ that corresponds to weight factor of the pocket coordinate loss, we vary it in {1.5, 10.0, 15.0}. The experimental results about the $α_3$ are shown as follows:
>
> | $\alpha_3$ | Ligand RMSD (Å) | | | Pocket RMSD (Å) | |
> |:-:|:-:|:-:|:-:|:-:|:-:|
> | | Mean $\downarrow$ | Median $\downarrow$ | < 2Å (%) $\uparrow$ | Mean $\downarrow$ | Median $\downarrow$ |
> | 1.5 | 5.22 | 3.05 | 39.27 | 1.50 | 0.89 |
> | 10.0 | 5.18 | 2.95 | 39.27 | 1.25 | 0.78 |
> | 15.0 | 5.44 | 2.96 | 40.59 | 1.10 | 0.63 |
>
> It can be observed that a larger $α_3$  is necessary to effectively improve pocket RMSD, because the initial AlphaFold2-predicted apo protein conformation is already quite similar to the ground-truth holo protein conformation. A smaller $α_3$ results in weaker gradients for the pocket coordinate loss within the multi-task learning framework, making it less effective in adjusting the pocket conformation. We finally set the $α_3$ as 15.0 to achieve improvement in both ligand and pocket structures.
>
> >**W14:** Reproducibility: Although the code is open source, the README section is empty, and usage instructions are not provided. If the README were clarified, I could test and reassess the code’s reproducibility. Additionally, the conda environment in the YAML file is named "FABind," raising anonymity concerns regarding double-blind review, as it suggests potential overlap with FABind’s authors.
>
> **A:** Thanks for your feedback on reproducibility. We have reorganized the code repository to enhance usability and reproducibility. A detailed README file has been added, which includes usage instructions, setup guidance, and example commands to run the code. We have also provided model checkpoints to facilitate easy replication of our results. Regarding the conda environment named as "FABind", it is because we use the same Python environment as FABind, thereby we use this name when we create the conda environment, which is not related to any author information. We will change it to FABFlex if you believe this is necessary.
>
> >**W15:** Minor Mistake, "Subsequent" is used twice: "The predicted pocket sites by pocket prediction module enable the subsequent subsequent ligand"
>
> **A:** Thanks for pointing out this minor mistake. We appreciate your meticulous review. We have corrected the repeated word "subsequent" in the revised version of the paper.
>
> >**Q4:** Did you experiment with different pocket site radii, such as 10 Å or 5 Å, in addition to 20 Å?
>
> **A:** Thanks for your valuable comments. We refer to the analysis presented in FABind+ paper [7] to explain the selection of 20 Å as the pocket radius. Referring to Figure 9 in FABind+ paper, the horizontal axis represents the distance between predicted pocket center and the farthest ground truth ligand atom, which reflects that the minimum pocket radius required from the predicted pocket center to cover all ligand atoms. It can be observed that for most data points in PDBBind, a pocket radius of 20 Å is appropriately suited to cover all ligand atoms. Thus, our choice of a 20 Å pocket radius is grounded in the analysis provided by previous work.
>
> [7] Gao K, Pei Q, Zhu J, et al. FABind+: Enhancing Molecular Docking through Improved Pocket Prediction and Pose Generation[J]. arXiv preprint arXiv:2403.20261, 2024.

---

> ### Author Response · Authors · 2024-11-22
> **Looking forward to your reply**
>
> Dear Reviewer H5gN,
>
> Thanks for your valuable time and great efforts in reviewing our work and providing insightful comments. We hope that our answers and general responses have helped to clarify the points discussed. Please let us know if there is anything else you require further information on or if there are any additional concerns you might have.
>
> Best regards,
>
> Authors of # 7007

---

> ### Author Response · Authors · 2024-11-25
> **Eagerly looking forward to your valuable feedback**
>
> Thank you for your valuable time and effort in reviewing our paper and providing insightful comments. With **only two days** remaining in the discussion period and having not yet received your feedback, we would like to kindly confirm whether our response has addressed all your concerns. We would be grateful for any additional discussion or suggestions.
>
> We deeply understand that reading others' papers and providing suggestions is a time-consuming and arduous task. Therefore, we sincerely appreciate every reviewer who thoughtfully reviewed and provided comments on our paper. It is your suggestions that help us  make our work better. **The paper you have read and the responses we have provided are the result of our authors' utmost effort, dedication, and hard work. It holds great significance for us, we sincerely look forward to discussing with you**.
>
> Thanks for your understanding. Best wish to you.
>
> Authors of # 7007

---

> > ### Comment · Reviewer_H5gN · 2024-11-25
> >
> > First of all, I appreciate the authors' comprehensive responses and the additional experiments they conducted. While their efforts are convincing to some extent, I am still concerned about the lack of benchmarking on a very basic method like PoseBuster. This omission raises doubts about the robustness of their benchmarking process. If the authors could demonstrate the percentage of ligands in the test set achieving RMSD < 2Å and 100% success in PoseBuster, I would be willing to reconsider my score. From what I can see, there doesn’t appear to be a significant improvement in RMSD compared to existing DiffDock results.
> >
> >
> > Additionally, one aspect that caught my attention is the lack of a detailed comparison between the two versions of DiffDock[1,2]. The authors have not provided a comprehensive evaluation of the runtime and performance differences between DiffDock-1[1] and DiffDock-2[2], as they have done with FABind. I would strongly recommend the authors include a more detailed comparison of DiffDock-1[1] and DiffDock-2[2] in their results tables.
> >
> >
> > **For Further Improvements:** For future work, I believe FABFlex’s efficiency could be further highlighted by benchmarking it with AF-3, Boltz-1, and Chai in docking scenarios. Demonstrating the framework’s performance in these contexts would significantly underscore its importance and versatility.
> >
> > - [1] Corso G, Stärk H, Jing B, et al. DiffDock: Diffusion Steps, Twists, and Turns for Molecular Docking.
> > - [2] Corso G, Deng A, Polizzi N, et al. Deep Confident Steps to New Pockets: Strategies for Docking Generalization

---

> > > ### Author Response · Authors · 2024-11-28
> > > **Response to reviewer H5gN (Part 1/2)**
> > >
> > > We deeply appreciate your thoughtful suggestions and valuable comments. We fully agree that incorporating PoseBuster [1] assessment and introducing more experiments of DiffDock-L [2] are beneficial to improving the quality of our paper. Following the reviewer's suggestions, we have supplemented the experimental results as follows:
> > >
> > > 1. Firstly, we would like to emphasize that the docking scenarios of DiffDock series and FABFlex are different, where the former **DiffDock series targets to rigid docking using the holo protein as input**, yet the latter **FABFlex takes apo protein (predicted by AlphaFold2) as input for the flexible docking**. It may not be fully reasonable to directly compare our flexible docking performance with those of DiffDock and DiffDock-L in rigid docking scenario. To clarify this point, we have added the experimental results of **DiffDock series performing under rigid and flexible docking scenarios respectively**. The results are shown as follows:
> > >
> > > |Methods / Ligand RMSD (Å)|Percentile 25% ↓|Percentile 50% ↓|Percentile 75% ↓|Mean ↓| < 2Å (%)↑| < 5Å (%)↑|
> > > |:-:|:-:|:-:|:-:|:-:|:-:|:-:|
> > > |DiffDock (rigid)|1.4|3.3|7.3|7.1|38.2|63.2|
> > > |DiffDock (flexible)|1.82|3.92|6.83|6.07|29.04|60.73|
> > > |DiffDock-L (rigid)|1.24|2.26|5.96|5.57|47.35|69.21|
> > > |DiffDock-L (flexible)|1.55|3.22|6.86|5.99|36.75|62.58|
> > > |FABFlex (flexible)|1.40|2.96|6.61|5.44|40.59|68.32|
> > >
> > > It can be observed that DiffDock-L is indeed a more powerful competitor compared to DiffDock, especially it achieves 47.35% and 36.75% of ligand RMSD < 2Å in rigid and flexible docking scenarios respectively, **yet it also has a performance degradation such as ligand RMSD < 2Å from 47.35% to 36.75% when applying from rigid docking to flexible docking**. As we shown in Appendix C.3 of our paper, FABind and FABind+ also have similar phenomenon when applying from rigid docking to flexible docking. We have supplemented the DiffDock-L as new baseline in Table 1 of our revised paper for comparison.
> > >
> > > 2. Furthermore, we fully agree that considering the physical validity of predicted ligand structures is important. PoseBuster test suite with its test set is a widely used benchmark. However, similar to the first point, **we find that the PoseBuster test set is designed to assess rigid docking scenario, rather than for flexible docking**. Specifically, for a test case, it provides a holo protein file "PDB_CCD_protein.pdb" and an apo ligand file "PDB_CCD_ligand_start_conf.sdf" as inputs, to predict the holo ligand structure, and given true holo ligand file "PDB_CCD_ligands.sdf" is used to evaluate the prediction. To evaluate flexible docking on the PoseBuster test set, **we would need to construct a flexible docking setting by using AlphaFold2 to obtain the apo protein and align it with its holo protein conformation**. This is a cost-intensive task, essentially creating a flexible version of the PoseBuster benchmark.

---

> ### Author Response · Authors · 2024-11-28
> **Response to reviewer H5gN (Part 2/2)**
>
> 3. Therefore, to assess the physical validity of generated ligand structures in flexible docking scenario, **we employ the PoseBuster test suite to evaluate the ligand structures generated by various competitors on the flexible PDBBind benchmark, aligned with already provided AlphaFold2-predicted apo proteins**. The experimental results are presented as follows:
>
> |Methods|RMSD < 2Å (%) ↑|PB-valid & RMSD < 2Å (%)↑|
> |:-|:-:|:-:|
> |FABind|22.52|1.32|
> |FABind+|34.11|8.94|
> |DiffDock-L|37.42|16.89|
> |DynamicBind|30.79|14.57|
> |FABFlex|39.40|13.91|
>
> Following PoseBuster, we call that molecule poses which pass all validity tests in PoseBuster test suites are "PB-valid". It can be observed that FABFlex achieves the highest RMSD < 2Å percentage among all competitors, demonstrating its superior accuracy in predicting docking poses close to the ground truth holo ligand. However, the PB-valid & RMSD < 2Å of FABind, FABind+, and FABFlex is lower than those of DiffDock-L and DynamicBind, reflecting the strength of diffusion-based models in generating physically valid molecular poses. This may be attributed to the strategy that diffusion-based models adjust molecular structures via translational, rotational, and torsional movements, reducing the degrees of freedom. Notably, FABFlex achieves a PB-valid & RMSD < 2Å score (13.91%) comparable to DynamicBind (14.57%), larger than poor FABind (1.32%) and FABind+ (8.94%), showcasing its balance in achieving both accuracy and physical validity. From these results, we are indeed fully motivated to continue improving our FABFlex to achieve better PB-valid results, e.g., ensuring validity as a constraint during training.
>
> 4. Additionally, to analyze the physical validity comprehensively, we plot a figure to showcase the validity rates of all aspects by PoseBuster test suite. This figure has been supplemented in the Figure 9 of our revised paper and uploaded on the link of anoymous repository (https://anonymous.4open.science/r/FABFlex-7007/figures/validity_rate_plot.png). It can be observed that a core limitation regarding the physical validity of deep learning-based docking models lies in the "minimum_distance_to_protein", that is, the distance between protein-ligand atom pairs is larger than 0.75 times the sum of the pairs van der Waals radii. All docking methods demonstrate a low validity rate in this aspect. Additionally, the validity of other critical factors, such as "tetrahedral chirality", "internal steric clash", "internal energy", and "volume overlap with the protein", also requires further improvement. Overall, ensuring the physical validity of predicted ligand structures still remains an open question.
>
> 5. Moreover, AlphaFold3 [3], Boltz-1 [4], and Chai [5] are all groundbreaking and influential works in the modeling of protein, DNA, RNA, ligand structures, and their interactions. Compared to them, our work just make a little contribution to the docking area. In the future, we will continue making research in the field of 3D biology, hoping that we can make such pioneering and impactful studies in one day.
>
> [1] Buttenschoen M, Morris G M, Deane C M. PoseBusters: AI-based docking methods fail to generate physically valid poses or generalise to novel sequences[J]. Chemical Science, 2024, 15(9): 3130-3139.
> [2] Corso G, Deng A, Polizzi N, et al. Deep Confident Steps to New Pockets: Strategies for Docking Generalization[C]//The Twelfth International Conference on Learning Representations, 2024.
> [3] Abramson J, Adler J, Dunger J, et al. Accurate structure prediction of biomolecular interactions with AlphaFold 3[J]. Nature, 2024: 1-3.
> [4] Wohlwend J, Corso G, Passaro S, et al. Boltz-1: Democratizing Biomolecular Interaction Modeling[J]. bioRxiv, 2024: 2024.11. 19.624167.
> [5] Chai Discovery, Boitreaud J, Dent J, et al. Chai-1: Decoding the molecular interactions of life[J]. bioRxiv, 2024: 2024.10. 10.615955.

---

> > ### Comment · Reviewer_H5gN · 2024-11-28
> >
> > I thank the authors for the detailed analysis they provided. While it seems that FABFlex still does not demonstrate a significant improvement, I appreciate the effort put into the extensive benchmarking. Based on this, I have decided to raise my score. I hope the authors can continue to refine FABFlex and explore more favorable structures to enhance its impact.

---

> > > ### Author Response · Authors · 2024-11-29
> > > **Thank you**
> > >
> > > Dear Reviewer H5gN,
> > >
> > > We are truly grateful for your thoughtful feedback and constructive comments, which have been immensely helpful! Based on your insightful suggestions, we have supplemented our paper with several important experiments, including clash score, binding affinity, DiffDock series, and PoseBuster. These additions have significantly enhanced the depth and quality of our work. Moving forward, we will continue to explore this field in greater depth in the future, building on these findings and further advancing our understanding.
> > >
> > > Thank you once again for your time and valuable contributions!
> > >
> > > Best regards,
> > > Authors of #7007

---

### Official Review · Reviewer_raSe · 2024-11-04

**Soundness:** 3
**Presentation:** 3
**Contribution:** 3
**Rating:** 8
**Confidence:** 4

**Summary:**

This paper addresses the challenging and realistic scenario of blind flexible molecular docking, where the binding sites are unknown and the proteins exhibit dynamic behavior during the docking process. This is a critical problem that reflects how molecules interact with proteins. The paper argues that current flexible docking methods, predominantly reliant on sampling strategies with diffusion models, suffer from significant inefficiencies. To overcome this drawback, the paper explores the potential of regression-based model in handling flexible docking. Utilizing AlphaFold2-predicted apo protein conformations, this paper proposes an end-to-end regression-based model named FABFlex, designed to achieve both fast computation and accurate docking performance in blind flexible docking scenarios. Experiments show that FABFlex not only significantly enhances the ligand structures and positively performs impacts on pocket conformations, but also substantially accelerates docking speed compared to the recent SOTA method, DynamicBind.

**Strengths:**

1.	The paper tackles the blind flexible molecular docking scenario, which is a more practical and crucial setting compared to many existing studies that focus on rigid docking, where proteins are assumed to be static during the docking process.
2.	The architecture of the proposed model is intuitive and easy to comprehend, with each module specifically designed to address a subtask of the blind flexible docking problem. It is easy to follow.
3.	The model significantly outperforms the SOTA docking methods, such as DynamicBind, DiffDock, and TankBind, on ligand structure predictions. Additionally, it operates much faster than recent SOTA flexible docking method, DynamicBind (approximately 208 times).
4.	This model maintains a robust generalization ability on those unseen proteins. The visualization of the iterative mechanism is very interesting, illustrating how the ligand is gradually docked from the apo to the holo state.

**Weaknesses:**

1.	It is unclear why the number of ligand-protein sample pairs of PDBBind v2020 used in this paper is smaller than that in the existing studies such as TankBind, FABind.
2.	It seems that FABFlex relies on FABind layer as the fundamental component to construct the model, but the details of how this layer is adapted for use in FABFlex are not clearly articulated.
3.	It seems that the model assumes that there is only a single binding pocket in a given ligand-protein pair, whereas in reality, there are possibly multiple potential binding pockets.

**Questions:**

1.	Does FABFlex’s docking result have atomic clash problem? If so, how to mitigate or resolve this problem?
2.	Can you provide more details of the construction and workflow of FABFlex?
3.	Can you use some way to intuitively demonstrate the pocket prediction performance?

---

> ### Author Response · Authors · 2024-11-21
> **Response to reviewer raSe (Part 1/3)**
>
> We appreciate your time and effort and would like to address the concerns.
> >**W1:** It is unclear why the number of ligand-protein sample pairs of PDBBind v2020 used in this paper is smaller than that in the existing studies such as TankBind, FABind.
>
> **A:**  Thanks for your careful comparison. We use the PDBBind v2020 dataset for our experiments, with details in Appendix B.1 and Table 4. The dataset is split in the same way as TankBind, FABind, and DynamicBind, using structures before 2019 for training and validation, and those after 2019 for testing. The data processing is identical to that in DynamicBind, meaning our processed dataset is the same as the one used in DynamicBind, where each protein is aligned with its corresponding AlphaFold-predicted structure. **The dataset in our paper is slightly smaller than that due to the removal of cases that fail to align with AlphaFold2 conformations.** However, all methods in this paper are evaluated on the same processed dataset, ensuring a fair comparison.
>
> >**W2:** It seems that FABFlex relies on FABind layer as the fundamental component to construct the model, but the details of how this layer is adapted for use in FABFlex are not clearly articulated.
>
> **A:** Thanks for your valuable comments. You are correct that the FABind layer serves as the fundamental E(3)-equivariant graph neural network for constructing the entire FABFlex model. FABFlex is composed of three modules: a pocket prediction module, a ligand docking module, and a pocket docking module. **All of these modules are constructed by stacking multiple FABind layers**, with the numbers of layers for each module being {1, 5, 5}, as detailed in Appendix B.3. We have reviesed the description of FABind layer in Section 3.1 for clarification. And a short introduction/summary of FABind layer is as follows:
>
> The FABind layer is a specialized E(3)-equivariant graph neural network designed to handle ligand-protein heterogeneous graphs. It includes three message passing steps for each layer: (1) independent message passing updates embeddings and coordinates within the ligand and protein; (2) cross-attention message passing facilitates information exchange between ligand and protein nodes. (3) interfacial message passing focuses on updating the embedding and coordinates on the contact surface. In summary, FABind layer is a geometric graph neural network to model ligand-protein graph.  In this work, we utilize the FABind layer as a fundamental component to construct our model. More details about the FABind layer computation can be found in FABind paper [1].
>
> [1] Pei Q, Gao K, Wu L, et al. FABind: fast and accurate protein-ligand binding[C]//Proceedings of the 37th International Conference on Neural Information Processing Systems. 2023: 55963-55980.
>
> >**W3:** It seems that the model assumes that there is only a single binding pocket in a given ligand-protein pair, whereas in reality, there are possibly multiple potential binding pockets.
>
> **A:** Thanks for your thoughtful question. We agree that there are possibly multiple potential binding pockets for a protein. However, our method primarily focuses on the binding pocket with highest probability, which is also aligned with the perspective of the PDBBind benchmark. PDBBind comprises a large collection of experimentally measured holo ligand-protein complexes, emphasizing the strongest natural binding pocket. Based on our statistics, approximately 18% of proteins in PDBBind are found to have multiple binding pockets. Additionally, many of these cases with multiple binding sites are **particularly in symmetric or multimeric proteins**, which present a more complicated scenario compared to existing single pocket setting. Extending existing methods or developing new approaches to address multi-pocket docking is indeed a promising direction for future research, and we have mentioned this point in Section 5 of our paper.

---

> ### Author Response · Authors · 2024-11-21
> **Response to reviewer raSe (Part 2/3)**
>
> >**Q1:** Does FABFlex’s docking result have atomic clash problem? If so, how to mitigate or resolve this problem?
>
> **A:** Thank you for your question regarding atomic clash. Yes, atomic clashes are indeed a common limitation of deep learning-based methods, such as TankBind, FABind, DiffDock, DynamicBind and our FABFlex, compared to traditional docking software like Vina, Gnina and Glide that strictly enforce Van der Waals forces. Following DynamicBind, we employ a clash score as the metric to evaluate the steric clashes. It is defined as the root-mean-square of the van der Waals overlaps for all atom pairs between the ligand and the protein where the distance is less than 4Å. The clash score is formulated as follows:
>
> $\text{clash score} = \sqrt{\frac{\sum_{i=0}^N \text{VdW overlop}^2}{N}}$,
>
> where $N$ is the number of atom pairs with distances considered. A lower clash score indicates fewer or less steric clashes. We supplement experimental results of the clash score across baselines as follows:
>
> | Methods   | Vina | Glide | Gnina | TankBind | FABind | FABind+ | DiffDock | DynamicBind | FABFlex |
> |---------------|:------:|:-------:|:-------:|:----------:|:--------:|:---------:|:----------:|:-------------:|:---------:|
> | **clash score $\downarrow$** | 0.02 | 0.08  | 0.05  | 0.41     | 0.51   | 0.45    | 0.33     | 0.27        | 0.37 |
>
> These results have been added in Appendix C.7 of the revised paper as a supplement. It can be observed that traditional docking software achieves significantly lower clash scores, whereas all deep learning-based docking methods exhibit higher clash scores. This reflects a core issue with deep learning approaches: despite their improved docking performance, they are prone to physical clash problems. To our knowledge, some studies have discussed this limitation, such as PoseBusters [2]. We think that incorporating physical and chemical constraints into model design and optimization may be an approach to mitigate this problem. This is a promising direction for future research to develop more powerful docking methods. We also acknowledge this limitation and its potential resolution in the discussion of Appendix D.
>
> [2] Buttenschoen M, Morris G M, Deane C M. PoseBusters: AI-based docking methods fail to generate physically valid poses or generalise to novel sequences[J]. Chemical Science, 2024, 15(9): 3130-3139.

---

> ### Author Response · Authors · 2024-11-21
> **Response to reviewer raSe (Part 3/3)**
>
> >**Q2:** Can you provide more details of the construction and workflow of FABFlex?
>
> **A:** Thanks for your kind question. FABFlex model consists of three modules: a pocket prediction module, a ligand docking module, and a pocket docking module, with each module based on the FABind layer. The aim of FABFlex is to predict the holo (bound) structures of ligand and pocket from given apo (unbound) states. We make further description of the workflow of FABFlex as follows:
> - Specifically, the workflow of FABFlex begins with two inputs: an apo ligand initialized using RDKit, and an apo protein predicted by AlphaFold2. The apo ligand is placed at the **center of the apo protein** to construct a ligand-protein graph, which is processed by the pocket prediction module to identify potential binding pocket residues on the protein. The output of this module includes the residues constituting the predicted pocket area, which provides crucial guidance for the subsequent docking modules.
> - Subsequently, the apo ligand is translated to the **center of the pocket** to construct the ligand-pocket graph. The ligand docking module and pocket docking module operate on the ligand-pocket graph to predict the holo structures of the ligand and pocket, respectively. An iterative update mechanism is employed to enable continuous coordinate refinement between the ligand and pocket docking modules.
> - The final outputs of FABFlex are the predicted holo structures of the ligand and pocket, representing their bound conformations. Section 3.2.4 in our paper introduces the pipeline of our FABFlex model, and the pseudo code is provided in Appendix A.2 for further clarification.
>
> >**Q3:** Can you use some way to intuitively demonstrate the pocket prediction performance?
>
> **A:** Thanks for your constructive question. Table 2 in our paper illustrates the performance of pocket prediction, which evaluates the predicted binding pocket sites from two perspective of residue classification and pocket center position with several metrics including accuracy (CLS ACC) to evaluate the residue binary classification, mean-absolute-error (MAE), root-mean-square-error (RMSE) and Euclidean distance (EucDist) to evaluate the position of predicted pocket center. From the table, it can be observed that FABFlex outperforms the widely used pocket detection tool P2Rank. For instance, FABFlex achieves lower MAE of 3.29 Å compared to P2Rank's 4.04 Å, a lower RMSE of 4.83 Å versus 5.69 Å. These results indicate the effectiveness of FABFlex in identifying pocket residues. Moreover, we have supplemented two additional case studies in Figure 12 in Appendix C.11 of our revised paper to intuitively showcase the pocket prediction results of FABFlex.

---

> ### Author Response · Authors · 2024-11-22
> **Looking forward to your reply**
>
> Dear Reviewer raSe,
>
> Thanks for your valuable time and great efforts in reviewing our work and for your insightful questions. We hope that our answers and general responses have helped to clarify the points discussed. Please let us know if there is anything else you require further information on or if there are any additional concerns you might have.
>
> Best regards,
>
> Authors of # 7007

---

> > ### Comment · Reviewer_raSe · 2024-11-25
> >
> > Thanks for the rebuttal. I am satisfied that the authors have adequately addressed my concerns. The experimental results are solid. I recognize the authors' respone and valuable contribution of FABFlex. Exploring the potential of regression-based paradigms in more complex docking scenarios could be an interesting direction for future research. I will raise my score accordingly.

---

> > > ### Author Response · Authors · 2024-11-25
> > > **Thank you**
> > >
> > > Dear Reviewer raSe,
> > >
> > > We sincerely appreciate your great support and insightful comments, which mean a lot to us! Moreover, following your valuable suggestions, we have made revisions to enhance the quality of our paper.
> > >
> > > Thank you once again for your valuable time and efforts!
> > >
> > > Best regards,
> > >
> > > Authors of # 7007

---

### Author Response · Authors · 2024-12-01
**General response**

Dear All Reviewers and AC,

We appreciate all your insightful suggestions and valuable comments, which are immensely helpful for us. During the rebuttal phase, we are pleased to receive numerous positive remarks, such as the acknowledgement of realistic significance of blind flexible docking scenario (raSe, H5gN), as well as the recognition of our contribution as the first work to explore the potentials of regression-based methods for flexible docking (H5gN, R36f). The reviewers also appreciate the clarity and well-written quality of our paper (H5gN, w792), as well as the strong and solid experimental results (raSe, H5gN, w792, R36f), particularly achievements in high computational efficiency and improved docking accuracy compared to state-of-the-art methods (raSe, w792, R36f).

More importantly, inspired by the reviewers' comments and the interaction, our manuscript has been continually improved regarding the unclear parts or experimental verification about some specific points. We carefully followed the reviewers' suggestions to include additional experiments in our revised paper. For your reference, we supplement the main additions in the following:

- **New baseline DiffDock-L:** supplemented in Table 1 and Figure 6.
- **Clash score evaluation:** supplemented in Table 9 and Appendix C.8.
- **Binding affinity assessment:** supplemented in Table 10 and Appendix C.9.
- **Analysis of number of samplings**: supplemented in Table 6 and Appendix C.2.
- **Assessment of PoseBuster test suite:** supplemented in Table 8, Figure 9, and Appendix C.6.
- **Performance on pocket-based flexible docking:** supplemented in Table 11 and Appendix C.10.
- **Additional case study of pocket prediction:** supplemented in Figure 13 and Appendix C.12.

Thank you once again for all reviewers' valuable time and efforts. Your thoughtful comments, no matter about the pros or the cons, all have been instrumental in improving the quality of our paper and have inspired us to continue advancing this research.

Best regards,

Authors of #7007

---

### Public Comment · ~Anton_Ber1 · 2025-04-29
**Unseen proteins results**

Dear authors,

In Table 1 you reported RMSD for complexes on unseen protein receptors, which was not included in PDBBind test split. Could you please clarify, which complexes from PDB (or maybe another data source) was used as unseen protein receptors?

---

> ### Public Comment · ~Zizhuo_Zhang1 · 2025-04-29
> **Response about Unseen Proteins**
>
> Thanks for your focus on our work.
>
> For PDBBind v2020 test set, there are some complexes with proteins that are unseen in training set. We call these complexes as test cases with unseen proteins. If you see our code, you can find a file named as "unseen_test_pdb.txt" in the directory of "baselines". This file records the PDB id of those test protein-ligand complexes with unseen proteins.

---

### Meta-Review · Area_Chair_Z72e · 2024-12-16

**Metareview:**

In this submission, the authors propose an effective and efficient blind ligand-protein docking method, which achieves encouraging performance. The proposed method formulates the blind docking problem as a multi-task learning problem, designing three modules to predict pockets, the holo-structures of ligands, and those of protein pockets, respectively, and learning them iteratively. The proposed method is reasonable. In my opinion, it demonstrates the potential of the "bottom-up" strategy in the blind docking problem. The authors resolved the reviewers' concerns successfully in the rebuttal phase, providing more explanations with additional experimental results. Taking all these into account, I decide to accept this work and suggest the authors further merge the content in the rebuttal and discussion phase into the final paper.

**Additional Comments On Reviewer Discussion:**

In the rebuttal and discussion phase, all four reviewers fully interacted with the authors. All of them were satisfied with the authors' feedback, and finally, scored the submission positively.

---

### Decision · Program_Chairs · 2025-01-22

Accept (Poster)